# Contribution of epigenetic landscapes and transcription factors to X-chromosome reactivation in the inner cell mass

Maud Borensztein [1,2], Ikuhiro Okamoto[1,3,4], Laurène Syx[1,5], Guillaume Guilbaud[6], Christel Picard[1], Katia Ancelin [1], Rafael Galupa [1], Patricia Diabangouaya[1], Nicolas Servant[5], Emmanuel Barillot [5], Azim Surani[2], Mitinori Saitou[3,4,7,8], Chong-Jian Chen[9], Konstantinos Anastassiadis[10] & Edith Heard[1]

X-chromosome inactivation is established during early development. In mice, transcriptional repression of the paternal X-chromosome (Xp) and enrichment in epigenetic marks such as H3K27me3 is achieved by the early blastocyst stage. X-chromosome inactivation is then reversed in the inner cell mass. The mechanisms underlying Xp reactivation remain enigmatic. Using in vivo single-cell approaches (allele-specific RNAseq, nascent RNA-fluorescent in situ hybridization and immunofluorescence), we show here that different genes are reactivated at different stages, with more slowly reactivated genes tending to be enriched in H3meK27. We further show that in UTX H3K27 histone demethylase mutant embryos, these genes are even more slowly reactivated, suggesting that these genes carry an epigenetic memory that may be actively lost. On the other hand, expression of rapidly reactivated genes may be driven by transcription factors. Thus, some X-linked genes have minimal epigenetic memory in the inner cell mass, whereas others may require active erasure of chromatin marks.

[1] Institut Curie, PSL Research University, CNRS UMR3215, INSERM U934, UPMC Paris-Sorbonne, 26 Rue d'Ulm, 75005 Paris, France. [2] Department of Physiology, Development and Neuroscience, Wellcome Trust Cancer Research UK Gurdon Institute, University of Cambridge, Tennis Court Road, Cambridge CB2 1QN, UK. [3] Department of Anatomy and Cell Biology, Graduate School of Medicine, Kyoto University, Yoshida-Konoe-cho, Sakyo-ku, Kyoto 606-8501, Japan. [4] JST, ERATO, Yoshida-Konoe-cho, Sakyo-ku, Kyoto 606-8501, Japan. [5] Institut Curie, PSL Research University, Mines Paris Tech, INSERM U900, 75005 Paris, France. [6] Medical Research Council Laboratory of Molecular Biology, Francis Crick Avenue, Cambridge CB2 0QH, UK. [7] Center for iPS Cell Research and Application, Kyoto University, 53 Kawahara-cho, Shogoin, Sakyo-ku, Kyoto 606-8507, Japan. [8] Institute for Integrated Cell-Material Sciences, Kyoto University, Yoshida-Ushinomiya-cho, Sakyo-ku, Kyoto 606-8501, Japan. [9] Annoroad Gene Technology Co., Ltd, Beijing 100176, China. [10] Biotechnology Center, Technische Universität Dresden, Tatzberg 47, 01307 Dresden, Germany. Maud Borensztein and Ikuhiro Okamoto contributed equally to this work. Correspondence and requests for materials should be addressed to E.H. (email: edith.heard@curie.fr)

In mammals, dosage compensation is achieved by inactivating one of the two X chromosomes during female embryogenesis[1]. In mice, X-chromosome inactivation (XCI) occurs in two waves. The first wave takes place during pre-implantation development and is imprinted, resulting in preferential inactivation of the paternal X (Xp) chromosome[2]. In the trophectoderm (TE) and the primitive endoderm (PrE), which contribute, respectively, to the placenta and yolk sac, silencing of the Xp is thought to be maintained[3,4]. In contrast, in the epiblast (Epi) precursor cells within the inner cell mass (ICM) of the blastocyst, the Xp is reactivated and a second wave of XCI, this time random, occurs shortly after[5,6].

Initiation of both imprinted and random XCI requires the Xist long-non-coding RNA that coats the future inactive X (Xi) chromosome in cis. The essential role of Xist in initiation of imprinted XCI has been recently highlighted in vivo[7,8]. Xist RNA coating is followed by gene silencing, and in previous studies, we have shown that different genes follow very different silencing kinetics[7,9]. Several epigenetic changes take place on the future Xi, including depletion of active chromatin marks (e.g., tri-methylation of histone H3 lysine 4 (H3K4me3), H3 and H4 acetylation), and recruitment of epigenetic modifiers such as polycomb repressive complexes PRC1 and PRC2, that result, respectively, in H2A ubiquitination and di-and tri-methylation of histone H3 lysine 27 (H3K27me3)[10]. The Xi is also enriched for

mono-methylation of histone H4 lysine K20, di-methylation of histone H3 lysine K9 and the histone variant macroH2A[5,6,11]. Only during random XCI, in the Epi, does DNA methylation of CpG islands occur to further lock in the silent state of X-linked genes, accounting for the highly stable inactive state of the Xi in the embryo-proper, unlike in the extra-embryonic tissues where the Xp is more labile[12–14].

Much less is known about how the inactive state of the Xp is reversed in the ICM of the blastocyst. X-chromosome reactivation is associated with loss of Xist coating and repressive epigenetic marks, such as H3K27me3[5,6]. Repression of Xist has been linked with pluripotency factors such as Nanog and Prdm14[15,16]. Studies on the reprogramming of somatic cells to induced pluripotency have shown that X-chromosome reactivation required Xist repression and that it occurs after pluripotency genes are expressed[17]. However, a previous study proposed that the reactivation of X-linked genes in the ICM operates independently of loss of Xist RNA and H3K27me3 based on nascent RNA-fluorescent in situ hybridization (FISH) and allele-specific reverse-transcribed polymerase chain reaction (RT-PCR) analysis of a few (7) X-linked genes[18]. Therefore, it is still unclear how X-chromosome reactivation in the ICM is achieved and whether it relies on pluripotency factors and/or on loss of epigenetic marks such as H3K27me3. Furthermore, whether loss of H3K27me3 is an active or a passive process has remained an open

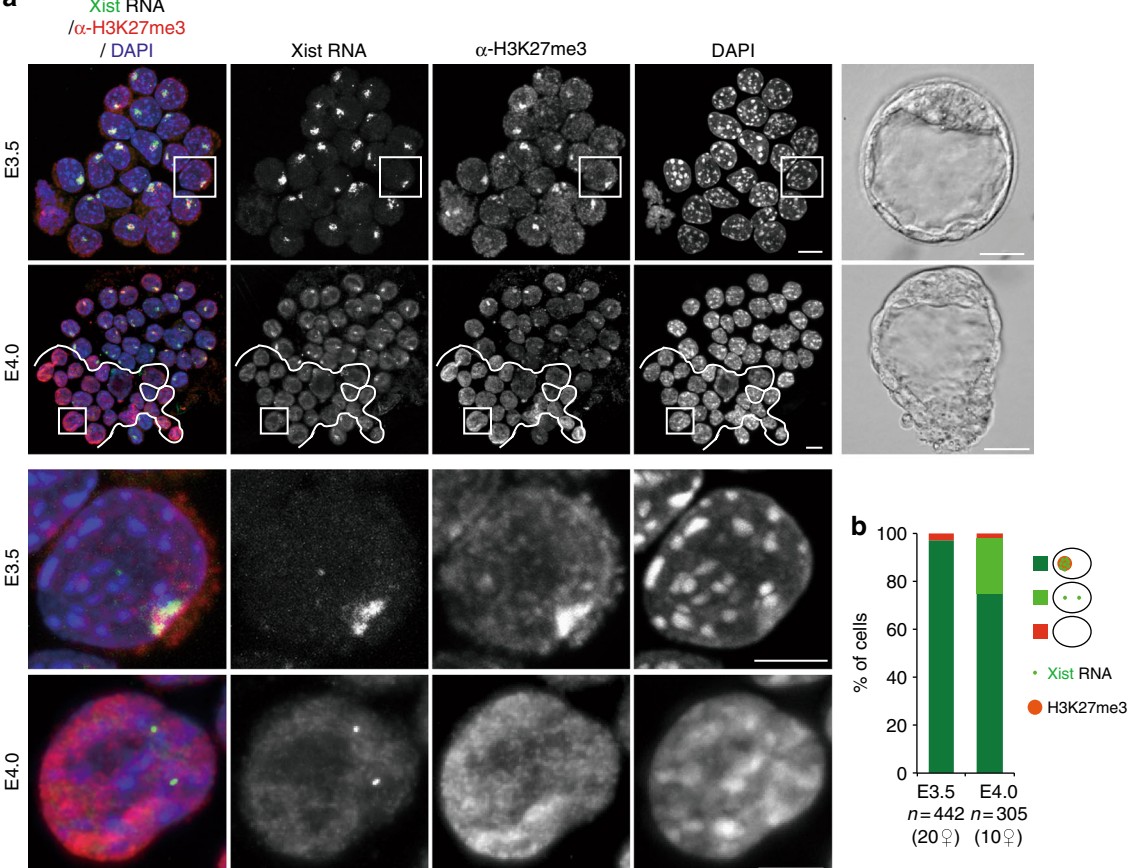

**Fig. 1** Xist RNA and H3K27me3 profiles in the ICM cells of early and mid blastocysts. **a** Examples of individual ICM of early (E3.5) and mid (E4.0) pre-implantation stage embryos (photographs right panel, scale bar 20 μm) analysed by immunolabelling with antibodies against H3K27 tri-methylation (red) combined with Xist RNA-FISH (green). For each stage, an intact ICM (scale bar, 10 μm) and an enlarged nucleus (scale bar, 5 μm) are shown (IF/RNA-FISH). The cells below the white line illustrate the cluster of cells that have lost Xist RNA coating and H3K27me3 enrichment on the Xp and are presumably the epiblast. **b** Proportion of ICM cells showing enrichment of H3K27me3 on the Xist RNA-coated X-chromosome in early and mid blastocyst stages are presented as mean (right panel). Below the graph, the total cell number analysed is indicated, followed by the number of female embryos analysed in brackets. *ICM* inner cell mass, *RNA-FISH* RNA-fluorescent in situ hybridization, *IF* immunofluorescence

question. Given the speed of H3K27me3 loss on the Xp from embryonic days 3.5 to 4.5 (E3.5–E4.5, i.e., 1–2 cell cycles), it is possible that active removal of the methylation mark might occur. Genome-wide removal of tri-methylation of H3K27 may be catalysed by the JmjC-domain demethylase proteins: UTX (encoded by the X-linked gene *Kdm6a*), UTY (a Y-linked gene) and JMJD3 (encoded by *Kdm6b*)[19–22]. Diverse roles have been proposed for these demethylases[23–25]. JMJD3 appears to inhibit reprogramming[26], whereas UTX plays a role in differentiation of the ectoderm and mesoderm[27] and has been proposed to promote

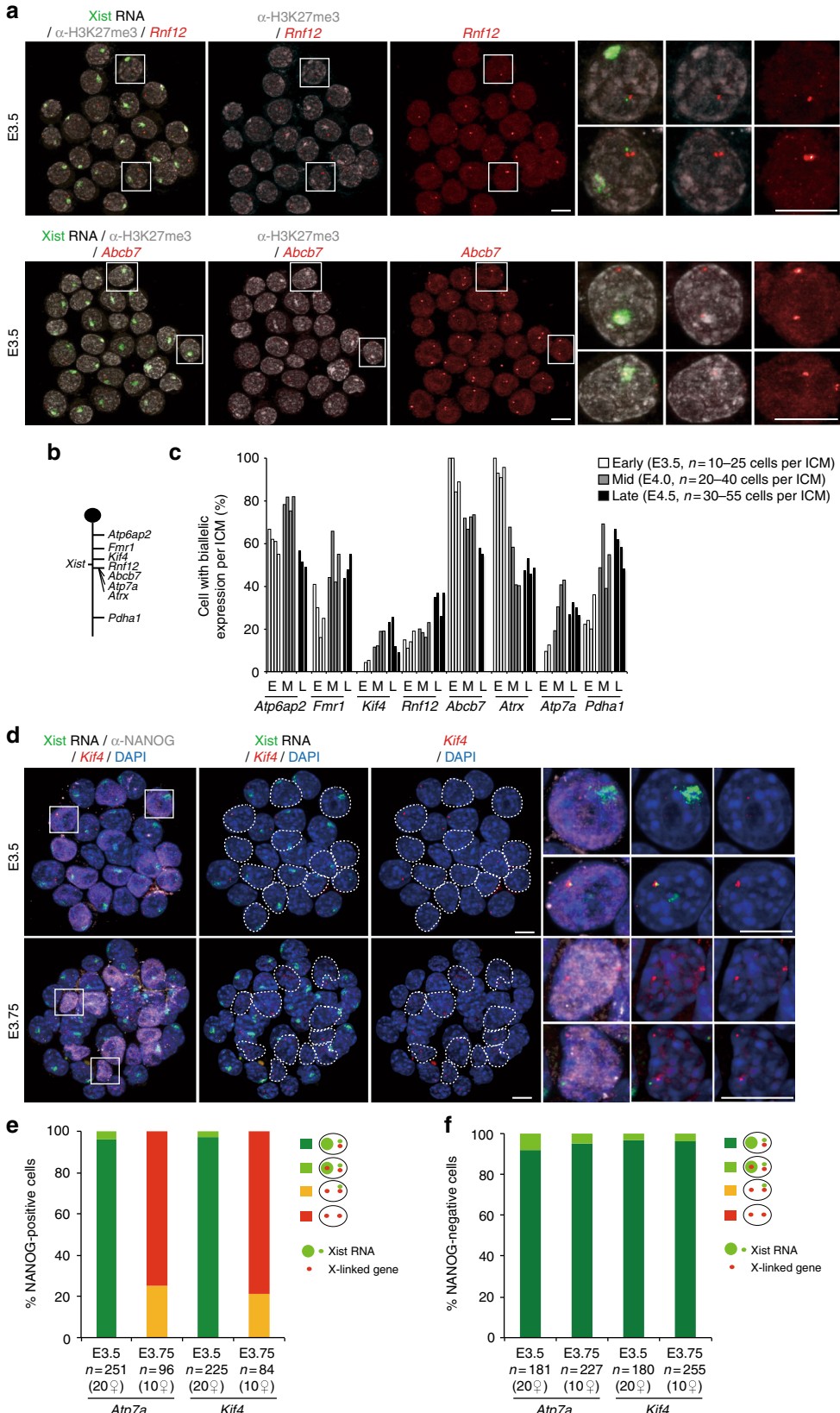

somatic and germ cell epigenetic reprogramming[24]. Interestingly, the *Utx* gene escapes XCI, being transcribed from both the active and inactive X chromosomes in females[28]. This raises the intriguing possibility that Utx might have a female-specific role in reprogramming the Xi in the ICM. *Utx* knockout mouse studies have suggested an important role of Utx during mouse embryogenesis and germ line development, but its exact molecular functions in X-linked gene transcriptional dynamics have not been assessed[21,22,24,29,30].

In this study, we set out to obtain an in-depth view of the nature of the X-chromosome reactivation process in the ICM in vivo. Here, we define the chromosome-wide timing of X-linked gene reactivation and examine what the underlying mechanisms might be both at the transcription factor and chromatin levels. This work points to distinct mechanisms at play for the reactivation of X-linked genes in the ICM, with broad implications for our understanding of epigenetic reprogramming in general.

## Results

**Chromatin dynamics of the paternal X-chromosome in the ICM.** Xp reactivation has been described to occur in the pre-Epi cells of blastocysts[5,6], but its exact timing and associated epigenetic changes are less clear. To determine the dynamics of chromatin changes in the ICM during E3.5 (early) to E4.0 (mid) development, we performed immunosurgery on blastocysts at different stages in order to destroy outer TE cells and specifically recover the ICMs. By combining immunofluorescence with Xist RNA-FISH, we analysed the enrichment of H3K27me3, as it is known to accumulate on the Xp shortly after the initiation of XCI, from E2.5[5]. As previously reported[5,6], H3K27m3 was found to be enriched on the Xist RNA-coated X-chromosome in almost all ICM cells of early blastocyst embryos (E3.5, 10–25 cells per ICM) (Fig. 1). Just half a day later, (E4.0, 20–40 cells per ICM), H3K27me3 enrichment and Xist RNA coating were lost in ~25% of cells within the ICM (Fig. 1). The cells that lost Xist RNA coating and H3K27me3 enrichment on the Xp at E4.0 were often clustered together in close proximity in the ICM, suggesting that they represent the pre-Epi population (Fig. 1a). Thus global loss of H3K27me3 enrichment on the Xp occurs with similar dynamics to loss of Xist RNA coating in a subpopulation of ICM cells, presumably the pre-Epi, between E3.5 and E4.0.

**Specific timing of gene reactivation and lineage specificity.** A previous report showed that reactivation of some X-linked genes seems to initiate despite the presence of Xist RNA coating and H3K27me3 enrichment of the Xp in ICM cells at E3.5 (early stage blastocysts)[18]. Combining RNA-FISH and anti-H3K27me3 immunofluorescence, we analysed expression of two of these genes, *Rnf12* and *Abcb7* that are repressed during imprinted XCI by E2.5[7,9,31]. Strikingly, *Abcb7* and *Rnf12* showed very different reactivation behaviours in the early E3.5 ICM (Fig. 2a). While

*Rnf12* exhibited low biallelic expression (<20% of ICM cells), suggesting its Xp silencing is maintained, *Abcb7* was biallelically expressed in almost all ICM cells, despite the presence of Xist RNA coating and H3K27me3 enrichment on the Xp (Fig. 2a–c). These results are only partially concordant with the previous report[18], however, the discrepancy for *Rnf12* may be due to differences in the exact stage of blastocyst development examined, or to the mouse strains (B6D2F1 and C57BL/6JxCAST/EiJ here, compared to CD-1 and CD-1xJF1[18]).

We examined further genes for their timing of Xp reactivation in the ICM. We performed RNA-FISH in pre-implantation (E3.5, early) through to peri-implantation (E4.5, late) blastocysts for eight X-linked genes together with Xist (Fig. 2b). The genes were chosen based on their known range of silencing kinetics during imprinted XCI in pre-implantation embryos, including genes silenced early (prior to E3.0 such as *Kif4*, *Rnf12*, *Atp7a*, *Atrx* and *Abcb7*), late (after E3.0, e.g., *Pdha1*, *Fmr1*) or that escape XCI (e.g., *Atp6ap2*)[7,9]. Among the candidates, *Rnf12*, *Atp7a*, *Abcb7* and *Pdha1* were all previously described as being reactivated at the mid blastocyst stage (E4.0)[18].

Increased frequencies of biallelic expression were observed for most genes from the E4.0 stage onward (*Fmr1*, *Kif4*, *Atp7a* and *Pdha1* and *Rnf12*), indicating that they have reactivated in a subset of ICM cells (presumably pre-Epi cells) (Fig. 2c). However, *Atrx* displayed biallelic expression as early as E3.5, similarly to *Abcb7* gene (Fig. 2a, c). Thus, reactivation of *Atrx* and *Abcb7* occurs in the early ICM cells prior to any lineage segregation between Epi and PrE[32–34]. Interestingly, from E4.0, a decrease in biallelic expression of these two genes was seen in 30–60% of ICM cells (Fig. 2c). This decrease may indicate that these genes are silenced again, presumably in future PrE cells. In the case of *Atp6ap2*, which is a gene that normally escapes from XCI, as expected, it was found biallelically expressed in 60–80% of ICM cells at all stages[9] (Fig. 2c). Taken together, our data suggests that the reactivation of X-linked genes occurs with very different timing during ICM differentiation. Furthermore, we find that a subset of genes may be reactivated early on, but then become rapidly silenced again in a subpopulation of cells, presumably destined to become PrE.

To examine whether biallelic expression of late-reactivated genes correlates with pre-Epi differentiation (and thus NANOG protein), we performed NANOG immunofluorescence combined with RNA-FISH for Xist and two such X-linked genes (*Kif4* and *Atp7a*) in ICM cells of early and mid blastocysts (Fig. 2d). As expected from our previous RNA-FISH (Fig. 2c), we found that cells mostly displayed monoallelic expression of these genes at E3.5. And this was the case in both NANOG-positive and -negative cells (Fig. 2d–f). *Kif4* and *Atp7a* then showed reactivation at E3.75 (Fig. 2c) and the biallelic cells are almost all NANOG positive (Fig. 2e). Moreover, biallelic expression of these genes was always observed in the absence of a Xist RNA

---

**Fig. 2** Xist RNA, X-linked gene expression and H3K27me3 profiles in the ICM cells of early to late blastocyst stage embryos. **a** Examples of individual ICM analysed by immunolabelling with antibodies against H3K27 tri-methylation (grey scale) and combined with RNA-FISH for Xist RNA (green) and primary transcription from the X-linked genes (red), together with representative nucleus are shown (scale bar, 10 μm). **b** Schematic representation of the X-chromosome showing the location of the loci analysed in **a** and **c**. *Atp6ap2* gene is known to escape XCI in 60–80% of blastocyst cells[9] and used as a control of the experiment. **c** Percentage (mean) of cells showing biallelic expression for X-linked genes in ICM of independent early (E3.5), mid (E4.0) and late (E4.5) blastocyst stage embryos. **d** Examples of individual ICM analysed by immunolabelling with NANOG (grey scale), combined with RNA-FISH for Xist (green) and X-linked genes (*Atp7a* and *Kif4*) (red) at early (E3.5) and mid (E3.75) blastocyst stage embryos. For each stage, an intact ICM (IF/RNA-FISH) and enlarged nuclei (white squares) are shown. Dotted lines indicate the position of NANOG-positive cells (scale bar, 10 μm). **e** Proportion (mean) of NANOG-positive ICM cells showing different Xist and X-linked gene expression patterns at early (E3.5) and mid (E3.75) blastocyst stage embryos. Below the graph, the total cell number analysed is indicated, followed by the total number of female embryos analysed in brackets. **f** Proportion (mean) of NANOG-negative ICM cells showing different *Xist* and X-linked gene expression patterns at early (E3.5) and mid (E3.75) blastocyst stage embryos. Below the graph, the total cell number analysed is indicated, followed by the total number of female embryos analysed in brackets

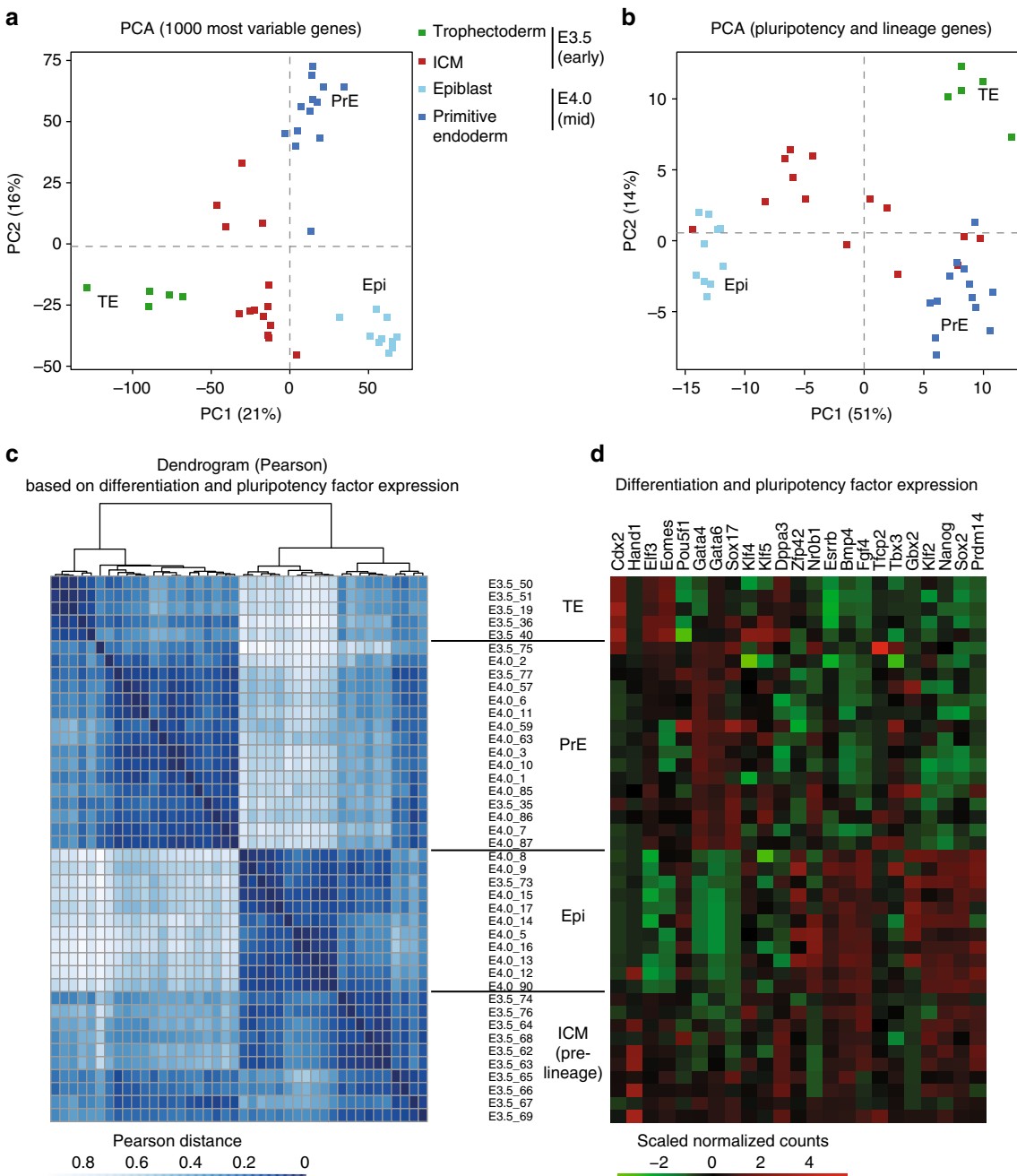

**Fig. 3** Single-cell RNAseq reveals loss of heterogeneity in the E4.0 mid ICM compared to early E3.5 ICM. Principal component analysis (PCA) based on scRNAseq data from trophectoderm (E3.5), early (E3.5, 10–25 cells per ICM) and mid (E4.0, 20–40 cells per ICM). ICM cells on the 1000 most variable genes (**a**) and on published pluripotency and differentiation candidate genes (n = 23, list in **d**) (**b**). Different stages are designed by different colours. n = 14, 23 and 5 cells, respectively, for E3.5 ICM, E4.0 ICM and E3.5 TE (details of each single cell is listed in Supplementary Data 1). **c** Hierarchical clustering (top) and Pearson distance (bottom) of pluripotency and lineage genes (listed in **d**) expression variation in E3.5 and E4.0 single cells, based on Pearson's correlation. Cells were clustered by lineage (TE, PrE and Epi), then by stage. n = 42 single-cell samples. **d** Level of expression of the 23 candidate genes involved in pluripotency and lineage differentiation in the 42 single-cell samples and used to classify cells according to their lineage are shown. Cells were ordered according to the hierarchical clustering in **c**. TE trophectoderm, PrE primitive endoderm, ICM inner cell mass, Epi epiblast

cloud in NANOG-positive cells (Fig. 2e, f). This corroborates previous observations that *Atp7a* is reactivated only in cells expressing Nanog[18]. Our results suggest that both Nanog expression and loss of Xist RNA coating are linked to biallelic expression of late-reactivated genes, but that Nanog expression alone is not sufficient. Taken together, our data point to reactivation in a lineage-specific manner beyond the mid ICM

stage for genes that are late-reactivated. They also reveal a lineage-independent reactivation of the early-reactivated genes at E3.5 ICM.

**scRNAseq of early and mid pre-implantation female ICMs.** The remarkable diversity in X-linked gene reactivation observed above (Fig. 2) prompted us to explore the Xp reactivation process on a

chromosome-wide scale. Furthermore, given the mixture of cells in the ICM, some of which are destined to become PrE, while others will become Epi, we were interested to know whether reactivation or silencing maintenance of Xp-linked genes correlated with expression of pluripotency factors (e.g., Nanog, Oct4, Sox2) and/or PrE factors (e.g., Gata4, Gata6) at the single-cell level[35]. We therefore performed single-cell RNA sequencing (scRNAseq) on ICMs of E3.5 and E4.0 pre-implantation female hybrid F1 embryos. We also used published ICM cells and TE cells where imprinted XCI is maintained[7]. The F1 hybrid blastocysts were derived from interspecific crosses between *Mus*

*musculus domesticus* (C57BL/6) females and *Mus musculus castaneus* (CAST/EiJ) males. Single cells from individual ICMs were collected and polyadenylated RNA amplified from each cell according to the protocol by Tang et al.[36] (Supplementary Data 1). We first assessed the extent to which transcriptomes of single cells from early (E3.5) and mid (E4.0) blastocysts were associated, using principal component analyses (PCA, Fig. 3a). We found that E3.5 ICM cells still showed substantial heterogeneity compared to E4.0 ICM single cells, which clustered into two distinct groups. Nevertheless, some signs that two subpopulations are emerging could be seen at E3.5. This revealed that lineage specification between PrE and Epi precursor cells could be important. We also performed PCA analyses (Fig. 3b) based on the expression levels of known pluripotency and differentiation factors, listed in Fig. 3d. As expected from previous studies, E4.0 ICM cells fall into two clearly separated groups, either PrE (expressing markers such as Gata4 and Gata6) or Epi (expressing markers such as Nanog and Prdm14)[32,33]. No strong association was observed in E3.5 ICM cells with the exception of a few cells ($n = 3$ pre-PrE and $n = 1$ pre-Epi at E3.5, Fig. 3b), supporting the idea that PrE and Epi lineages begin to be specified, but are still not clearly established at the transcriptional level in E3.5 ICMs, as previously reported[34]. Next, we performed a correlation analysis based on the expression status of pluripotency and differentiation factors (Fig. 3c, d). We classified cells according to their developmental stage and pluripotency/differentiation factor status: E3.5_TE (TE of early blastocysts), E3.5_ICM (PrE-lineage ICM of early blastocysts), E4.0_PrE (PrE precursor cells of mid blastocysts) and E4.0_Epi (Epi precursor cells of mid blastocysts). This clearly supports a shift from still rather heterogeneous transcriptomes in E3.5 ICM cells, into two well-defined subpopulations of pre-Epi and PrE cells in E4.0 ICMs.

**Specific X-linked gene behaviour during reactivation.** We next investigated chromosome-wide X-linked gene activity between early (E3.5) and mid (E4.0) ICMs. To assess the parental origin of transcripts, we took advantage of the high rate of polymorphisms between C57BL6/J (maternal) and CAST/EiJ (paternal) genomes that enabled us to distinguish parental origin for informative transcripts (see Methods section). An in vivo heatmap of X-linked gene activity was generated for early (E3.5_ICM) and mid (E4.0_PrE and E4.0_Epi) blastocyst stages and compared to female TE cells at E3.5 (TE) as controls of XpCI maintenance (Fig. 4 and Supplementary Fig. 1). Control TE cells displayed 21% (18 out of 86) of biallelically expressed genes (allelic ratio >0.2, Fig. 4), and 17 of these genes are well-known escapees[7]. Interestingly, E3.5 ICM cells showed a higher number of biallelically expressed genes when compared to TE. We found that 51% of X-

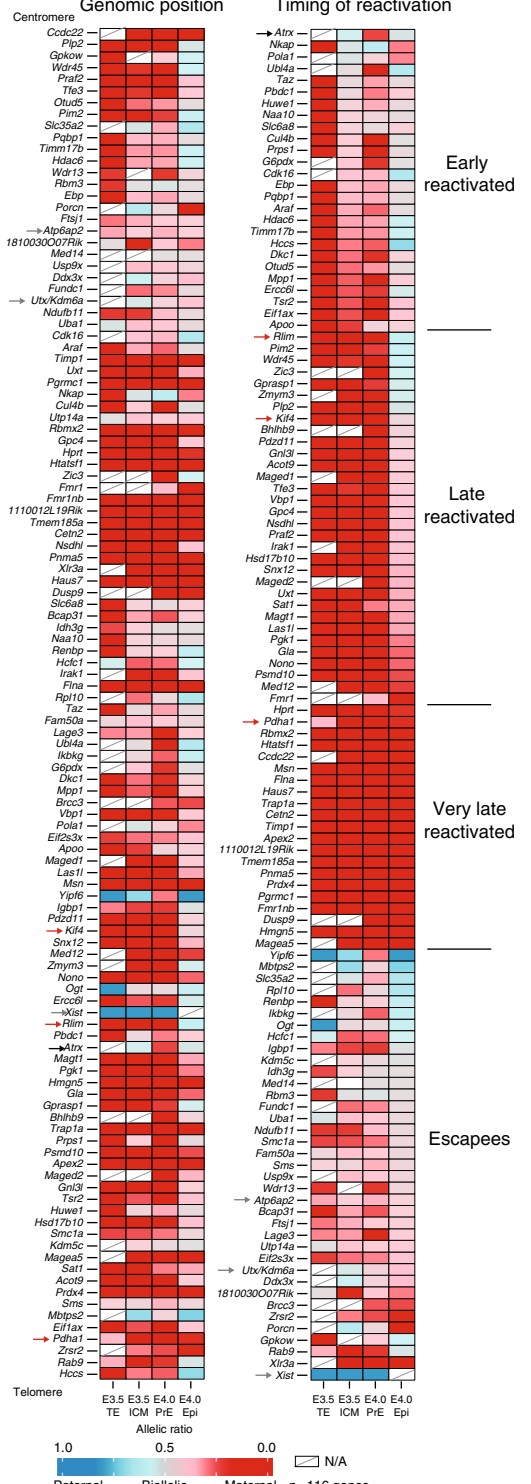

**Fig. 4** Different X-linked gene reactivation behaviours in the ICM. The mean of allele-specific expression ratios for each informative and expressed X-linked gene in E3.5 (trophectoderm and ICM) and E4.0 (primitive endoderm and epiblast) female C57BL/6JxCAST/EiJ embryos are represented as heatmaps, with strictly maternal expression (ratio ≤0.15) in red and strictly paternal expression (ratio ≥0.85) in blue. Colour gradients are used in between these two values as shown in the key. Genes are ordered by genomic position (left) or by timing of reactivation (right). Further information is provided in Supplementary Data 2 and Methods section. Black, red and grey arrows are, respectively, highlighting example of early-, later-reactivated genes and escapees. As expected, Xist RNA is paternally expressed in the trophectoderm cells. *Ogt* and *Yipf6* genes display similar paternal expression in the trophectoderm, escape imprinted XCI, and show random monoallelic expression and CAST/EiJ bias, respectively, (Supplementary Fig. 1)[7]. $n = 116$ genes, NA, data not available (below threshold)

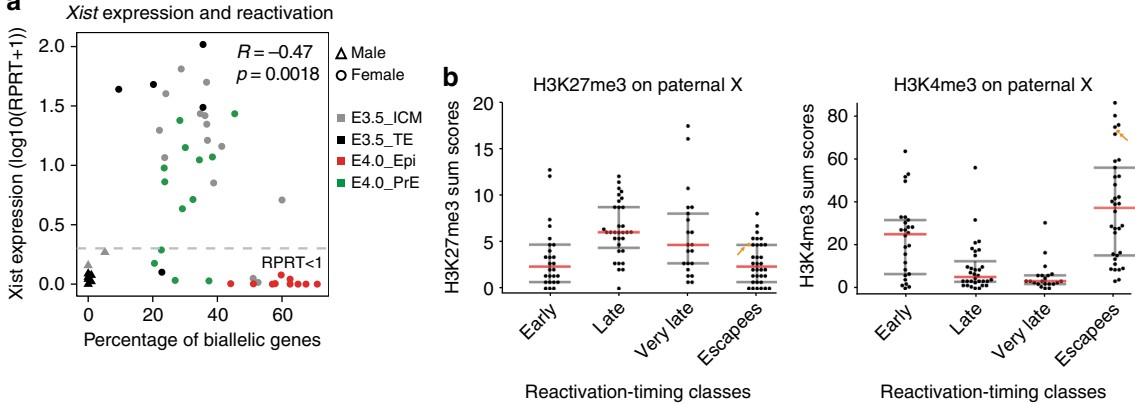

**Fig. 5** Link between Xist expression, epigenetic landscapes and Xp reactivation. **a** Anti-correlation is shown between the level of Xist expression and the number of biallelically/reactivated and informative X-linked genes in scRNAseq (Spearman correlation). Male E3.5 single cells have been added and used as control for Xist expression and X-linked gene parental expression. Genes with level of expression as (RPRT <1) are considered as non-expressed in our samples. **b** Enrichment of H3K27me3 and H3K4me3 on paternal X-chromosome obtained from Zheng et al.[38] shows significant differences (by Wilcoxon test) between early and escapee reactivation-timing classes compared to late and very late. Low cell chromatin immunoprecipitation sequencing (ChIPseq) have been performed with ICM cells of pre-implantation embryos (pooled between E3.5–E4.0) after immunosurgery of the ICM[38]. Activated genes show an excess of H3K4me3 and repressed ones an enrichment of H3K27me3. Xist is highlighted with an orange arrow. Early vs. late ($p = 2.29 \times 10^{-4}$ for H3K27me3 and $p = 1.63 \times 10^{-3}$ for H3K4me3) and very late ($p = 2.51 \times 10^{-2}$ for H3K27me3 and $p = 3.95 \times 10^{-4}$ for H3K4me3) and escapees vs. late ($p = 1.95 \times 10^{-6}$ for H3K27me3 and $p = 2.09 \times 10^{-7}$ for H3K4me3) and very late ($p = 7.33 \times 10^{-3}$ for H3K27me3 and $p = 6.73 \times 10^{-8}$ for H3K4me3) by Wilcoxon test

linked genes were expressed from both X-chromosomes in E3.5 ICM (55 biallelic genes out of 107, e.g., Atrx), despite the sustained expression of Xist. This supports our findings based on RNA-FISH for early-reactivated genes (Fig. 2) and further reveals the scale of such early reactivation. Intriguingly, several of these reactivated genes (e.g., Atrx, Ubl4a and Eif1ax) are rapidly silenced again half a day later in PrE precursor cells only, as defined by the expression of 23 differentiation and pluripotency markers (e.g., Gata4, Gata6 and Nanog, Fig. 3) (Fig. 4, right). These data suggest that oscillations in the expression states of some genes on the Xp (such as Atrx) occur within a subpopulation of ICM cells that will give rise to the PrE, where XpCI persists ultimately[4]. Our RNA-FISH data confirms that Atrx is transiently expressed from both X chromosomes even in the cells that will give rise to the PrE as it is found biallelically expressed in 90–100% of early ICM cells (Fig. 2c). Our results reveal that there may be fluctuations in the inactive state of some Xp-linked genes during ICM progression, in the precursor cells of the PrE, rather than a straightforward maintenance of Xp silencing as previously thought.

In Epi precursor cells, based on pluripotency factor expression (Fig. 3), at E4.0, we noted an absence of Xist expression and a marked progression in Xp-chromosome reactivation as 77% of genes become biallelically expressed (89 out of 115, Fig. 4). Genes that showed reactivation only in Epi precursor cells were classified as late-reactivated genes (Fig. 4 and Supplementary Data 2) and confirmed our previous RNA-FISH data (e.g., Rnf12, Kif4, Fig. 2). Interestingly, some genes, classified as 'very late reactivated', still appear to be repressed on the Xp, even at E4.0. In the case of Pdha1, this gene was found to be reactivated in about 40% of ICM cells at E4.0 by RNA-FISH (Fig. 2c), compared to 4% of paternal expression in PrE and 18% in Epi, by scRNAseq (Supplementary Data 2). This could be explained by differences between nascent (RNA-FISH) and mature RNA (scRNAseq) for this gene, if the levels of paternal messenger RNA (mRNA) are not yet high enough for scRNAseq detection even though the gene has begun to be transcribed.

The above data show that X-chromosome reactivation can initiate for some genes independently of loss of Xist RNA coating

and before lineage segregation at E3.5 (Figs. 2a, c and 4). However, in the Epi precursor cells at E4.0, a higher percentage of biallelic X-linked genes was always observed in absence of Xist RNA (Fig. 5a). Indeed Xist expression level and the percentage of biallelically expressed X-linked genes were anti-correlated ($R = -0.47$, $p = 0.0018$, Spearman correlation). Taken together, our data suggest that some genes ($n = 26$ out of 116) undergo X-chromosome reactivation independently of Xist RNA and H3K27me3 loss, and that their expression fluctuates between early and mid ICMs, with many of them being re-silenced in the PrE lineage. On the other hand, the majority of X-linked genes become reactivated later (E4.0), solely in Epi precursor cells in which pluripotency factors such as Nanog are expressed, and Xist RNA and H3K27me3 enrichment are lost (Figs. 2, 4 and 5).

**Differential TF binding and H3K27me3 enrichment on the Xp.** Next, we set out to identify the features that are associated with the different categories of genes along the X as defined by their reactivation kinetics (early, late and very late or escapees). First, we assessed whether the timing of reactivation could be linked to the kinetics or efficiency of silencing of a particular gene[7]. Correspondence analysis revealed that kinetics of X-linked gene reactivation does not mirror the kinetics of silencing (Supplementary Fig. 2a). Timing of reactivation is not simply about the lapse of time since silencing was initiated, nor about the location of a gene along the X chromosome (Supplementary Fig. 2b). Although a slight tendency was observed for late- and very late-reactivated genes to be in close proximity of the Xist locus, Atrx and Abcb7 genes are both silenced early, lie close to the Xist genomic locus and yet are also reactivated early[7,9,31]. Furthermore, our previous work revealed that although early silenced genes preferentially lie inside the first Xist 'entry' sites as defined by Engreitz et al. in embryonic stem cells (ESCs), the late- and very late-reactivated genes failed to show any significant correlation with Xist entry sites (Supplementary Fig. 2c)[7,37]. Gene expression level was also not found as an obvious predictor of early or late reactivation (Supplementary Fig. 2d). We thus hypothesize that late- and very late-reactivated genes may have acquired an epigenetic signature that prevents their rapid

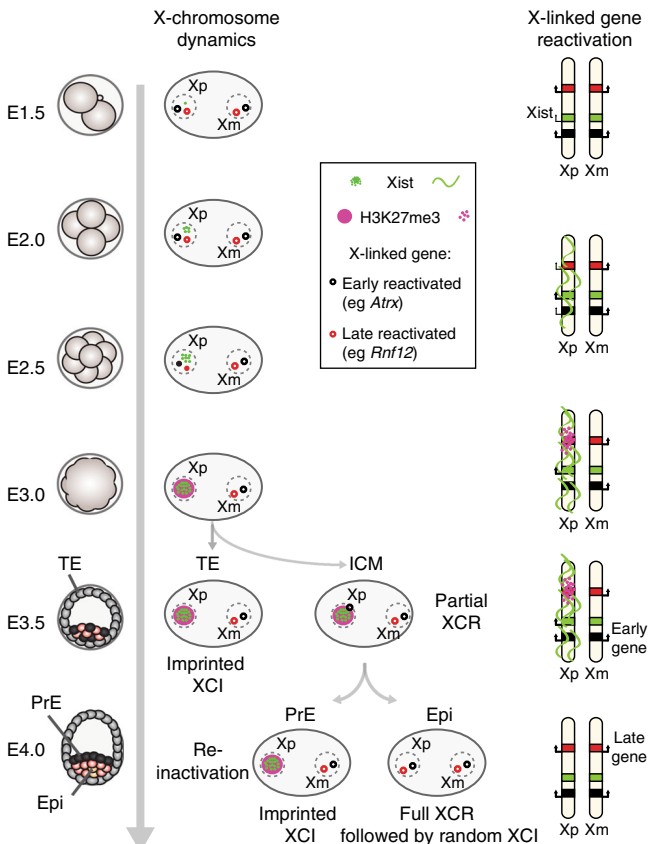

**X-chromosome dynamics**

E1.5

E2.0

E2.5

E3.0

E3.5 — TE

E4.0 — PrE / Epi

TE / ICM

Imprinted XCI

Partial XCR

Re-inactivation

PrE — Imprinted XCI

Epi — Full XCR followed by random XCR

**X-linked gene reactivation**

Legend: Xist ; H3K27me3 ; X-linked gene: Early reactivated (eg *Atrx*) ; Late reactivated (eg *Rnf12*)

Early gene ; Late gene

**Fig. 6** Representation of Xp reactivation and different X-linked gene behaviours. Scheme of imprinted XCI, followed by reactivation in the ICM of the blastocyst. Xp silencing is triggered by the long non-coding Xist RNA, followed by H3K27me3 recruitment. At the early blastocyst stage (E3.5), imprinted Xp is maintained in TE, when some genes are already showing reactivation in the ICM, independently of Xist (early-reactivated genes). Those early genes are lowly enriched in H3K27me3 marks and highly enriched in H3K4me3 on their paternal allele compared to the later reactivated ones. A few hours later, when ICM cells are divided into PrE and Epi cells, Xp reactivation appears to be nearly complete only in the future epiblast cells, (based on loss of Xist coating and H3K27me3). In PrE, some early-reactivated genes have already been silenced again. This suggests a fluctuation of early-reactivated genes with different epigenetic memory requirements between early- and late-reactivated genes. *TE* trophectoderm, *PrE* primitive endoderm, *ICM* inner cell mass, *Epi* epiblast

reactivation in early ICM. Early-reactivated genes on the other hand might become expressed more rapidly due to specific transcription factors (TFs) overriding their silent state.

To test these hypotheses, we first examined recent allele-specific ChIPseq data for H3K27me3 and H3K4me3 in ICMs (pooled between E3.5–E4.0)[38,39]. We compared enrichment for H3K27me3 (left) and H3K4me3 (right) across their transcription start site (TSS) in our different reactivation-timing groups (Fig. 5b). We found a clear and significant enrichment of H3K27me3 on the Xp but not on the Xm of late- and very late-reactivated genes compared to early-reactivated genes (respectively $p = 2.29 \times 10^{-4}$ and $p = 2.51 \times 10^{-2}$ by Wilcoxon test) and escapees (respectively $p = 1.95 \times 10^{-6}$ and $p = 7.33 \times 10^{-3}$ by Wilcoxon test) (Fig. 5b, left and Supplementary Fig. 2e, left). Moreover, early-reactivated genes and escapees are statistically enriched in the H3K4me3 histone mark compared to late (respectively $p = 1.62 \times 10^{-3}$ and $p = 2.09 \times 10^{-7}$) and very late genes (respectively $p = 3.95 \times 10^{-4}$ and $p = 6.73 \times 10^{-8}$) (Fig. 5b,

right and Supplementary Fig. 2e, right). As expected, we confirmed that H3K4me3-highly enriched genes are globally more highly expressed than lowly enriched genes (Supplementary Fig. 2f). However, as no association was found between a high level of expression and early reactivation (Supplementary Fig. 2d), we hypothesize that paternal enrichment of H3K4me3 could simply be a consequence of biallelic expression of early-reactivated genes. Altogether, this reveals that different groups of genes display rather divergent histone marks at the time of X-chromosome reactivation (Fig. 6).

To explore the second hypothesis, that some TFs, including pluripotency factors, might drive expression from the Xp of early-reactivated genes, we first analysed the correlation or anti-correlation between gene expression genome-wide and the degree of X-linked gene reactivation in female single cells (Supplementary Data 3, see Methods section). As expected based on previous observations, Xp chromosome reactivation correlates with expression of pluripotency factors (e.g., Esrrb, Sox2, Nanog, Oct4 and Prdm14) and anti-correlates with PrE differentiation factors, such as Gata4, Sox17 and Gata6[5,6,15,16]. Intriguingly, a gene ontology analysis of the top correlated genes ($q$ values $< 0.005$) revealed that epigenetic modifiers are over represented (Supplementary Fig. 2g). This suggests that they may play a role in reprogramming epigenetic landscapes, including some loci on the Xp. This hypothesis will be further explored later.

We next examined previously published data sets of TF-binding sites in mouse ESCs and their TF-gene-associated score as calculated by Chen et al.[40] (see Methods section). In particular, we analysed the occurrence of fixation sites at X-linked genes for pluripotency factors involved in the Epi or mouse embryonic stem cell (mESC) state (Nanog, Esrrb, Klf4, Oct4, Sox2, Tcfcp211). Half of the X-linked genes, independent of their kinetics of reactivation and including escapees, presented at least one binding site for these pluripotency factors (Supplementary Fig. 3a). Their expression might be partially regulated by these factors[15,16], but the binding of these factors alone cannot explain the behaviour of early-reactivated genes. We next analysed Myc family-binding sites, as Myc expression was also found associated with X-chromosome reactivation, though to a lesser degree than pluripotency factors (Supplementary Data 3). Myc factors are expressed in early and mid ICM cells and there is a slight but significant association between high expression of *Myc* and *Mycl* genes and high rate of X-linked gene reactivation (Supplementary Fig. 3b). We therefore analysed for the presence of Myc family-binding sites (Myc- and Mycn-binding sites from Chen et al.[40]). Both escapees and early-reactivated genes showed a similar enrichment for Myc factor-binding sites compared to other X-linked and autosomal genes (Supplementary Fig. 3c). In comparison, late- and very late-reactivated genes were significantly depleted in Myc-binding sites, when compared to early-reactivated genes, escapees but also genome-wide ($p < 0.0001$ by Kruskal–Wallis). Myc TFs play a role in induced pluripotent stem cell (iPS) reprogramming[41] and have also been linked with a hypertranscribed state, described in ESCs and Epi[42]. Thus, early-reactivated X-linked genes and escapees may well be more efficiently targeted for reactivation on the silenced paternal X by the Myc TF family in early ICM, compared to late and very late genes that have fewer Myc-binding sites and would therefore be less responsive to these TFs.

In conclusion, the early reactivation of some X-linked genes, even prior to global loss of Xist RNA coating and H3K27me3 enrichment at E3.5, may be partly due to transcriptional activation by the Myc TF family, a local lack of H3K27me3 and an enrichment of H3K4me3. On the other hand, the majority of genes that are reactivated later show reduced numbers of Myc-binding sites as well as higher H3K27me3 enrichment and lower

H3K4me3. This suggests there may be differences between early- and late-reactivated genes in epigenetic memory states and responsiveness to some TFs.

**Involvement of UTX in efficient reactivation of late genes**. The above findings (Figs. 2 and 5) support a dependency between late-

and very late-reactivated genes on loss of Xist and H3K27me3 enrichment from the Xp. To explore the hypothesis that epigenetic marking via H3K27me3 might play a role in the resistance of some genes to early Xp reactivation, we decided to try an impair H3K27me3 removal during the reactivation process. To do so, we produced peri-implantation (E4.5, $n = 30$–55 cells per

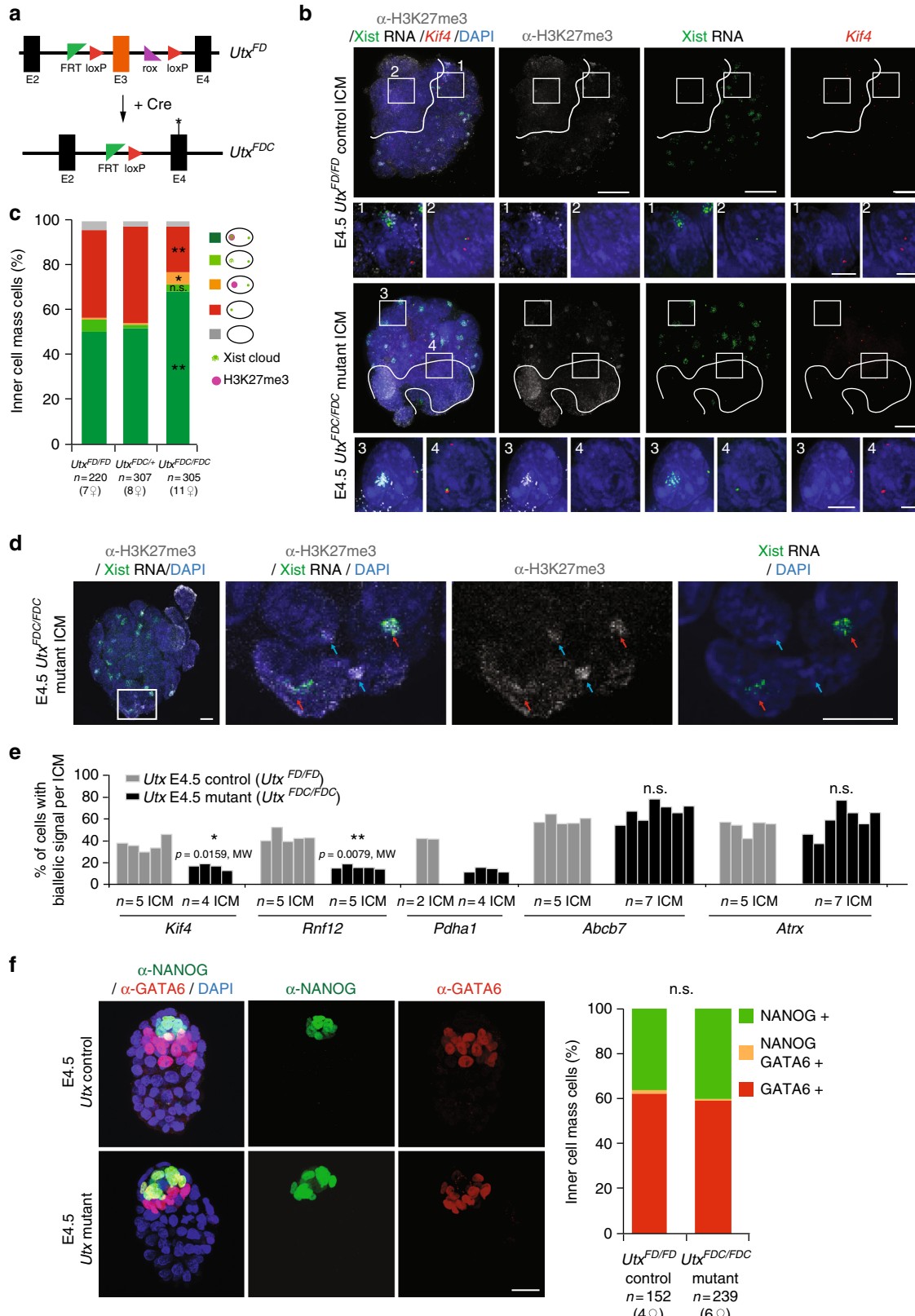

ICM) embryos lacking the X-linked histone demethylase UTX, which is reported to be specific for H3K27 demethylation[20–22] and may promote reprogramming[24]. Interestingly *Utx* is expressed in pre-implantation embryos and its expression remains high in early and mid ICM when it is down-regulated in TE, in which Xp inactivation is maintained (Supplementary Fig. 4a). Homozygous knockout *Utx^{FDC/FDC}* female embryos were obtained after matings between *Utx^{FDC/Y}* studs (knockout males) and *Utx^{FD/+}*; or *Utx^{FD/FD};GDF-9iCre* females (Fig. 7a) (the GDF-9-driven Cre enables efficient recombination in the maternal germ line)[30]. Absence of UTX protein was validated by immunofluorescence at late blastocyst stage (E4.5) (Supplementary Fig. 4b). Our aim was to assess if loss of Utx correlates with a prolonged accumulation of H3K27me3 at the inactive X, and its impact on the transcriptional status of late- or early-reactivated genes. We performed immunostainings on E4.5 control, heterozygous and mutant female ICMs (Fig. 7b, c). H3K27me3 enrichment on the Xp was retained in significantly more cells in *Utx* mutants compared to controls or heterozygous (respectively 73% vs. 50% and 52%, $p = 0.0002$, Kruskal–Wallis (KW) test). Furthermore a significantly higher proportion of Xist-negative cells with a H3K27me3 enrichment was found in the mutant (5.1% vs. 0.7% in controls, $p = 0.0067$, KW test, Fig. 7c, d). Altogether, our results are supportive of a scenario whereby UTX might be actively involved in removal of H3K27me3 from the Xp, following Xist downregulation. When X-linked gene expression was assessed in the *Utx* mutants, biallelic expression could be found with a corresponding absence of Xist, even in cells remaining H3K27me3-positive (Supplementary Fig. 4c, d). To explore the impact on gene reactivation further, we performed RNA-FISH on several late- or very late-reactivated genes (*Kif4*, *Rnf12* and *Pdha1*), as well as on early-reactivated genes (*Abcb7* and *Atrx*). Strikingly, late-gene reactivation was always lower in the mutant E4.5 ICMs compared to controls (about 50% decrease in *Utx* mutants, Fig. 7e). Furthermore, this decrease correlated with the increase in H3K27me3-positive cells in mutants. On the other hand, *Abcb7* and *Atrx* reactivation rates did not appear to be affected in *Utx* mutants. Thus early-reactivated genes do not appear to be sensitive to the lack of UTX and increase of H3K27me3 in ICM, supporting their H3K27me3-independent reactivation mechanism, as suggested by their depletion in local H3K27me3 enrichment (Fig. 5b).

Finally, to exclude the possibility that the apparent interference with X-chromosome reactivation might actually be due to delayed or abnormal development of ICM in *Utx* mutant embryos, leading to an increased proportion of PrE cells, we stained for NANOG (Epi marker) and GATA6 (PrE marker) in both control and mutant E4.5 female ICMs (Fig. 7f). No difference was seen in the total number of cells per ICM (with a mean of 38, 40 and 40 cells per ICM, respectively, for *Utx^{FD/FD}*, *Utx^{FDC/+}* and *Utx^{FDC/FDC}*) and in the proportions of NANOG (Epi) and GATA6-positive (PrE) cells. Thus, the absence of UTX did not impact on ICM progression, but does impair the efficiency of X-linked gene reactivation in vivo, at least for late-reactivated genes. In summary, our results reveal the existence of different epigenetic memory states during XpCI, with genes that tend to be enriched in H3K27me3 being more sensitive to the requirement for Utx for removal of H3K27me3 and reactivation, than others that are reactivated independently of global Xist and H3K27me3 enrichment.

## Discussion

Transcriptional reactivation of the Xp occurs in the mouse ICM during pre- to peri-implantation development. The extent and nature of this reprogramming process has remained poorly defined until now. Our single-cell analysis of Xp reactivation provides a chromosome-wide map of X-linked gene activity and strong evidence for multiple mechanisms involved in the loss of silencing of X-linked genes. Emergence of ICM, at the blastocyst stage, is a key event during early development. We now know that pluripotency factors such as Nanog will be retained in the Epi precursor cells that will give rise to the embryo-proper and this is where Xp reactivation occurs[5,6,33]. At the early blastocyst stage (E3.5), PrE and Epi precursor cells only begin to segregate and heterogeneity in the expression of specific lineage markers is still noticeable as confirmed in our study (Fig. 3). This initially high degree of cell-to-cell variation in pluripotency and lineage factor expression is lost by E4.0, when two transcriptionally distinct populations of cells can be observed. The pre-Epi cells are characterized by pluripotency genes and loss of *Xist* expression; PrE cells show Xist expression, decreased pluripotency gene expression and enhanced lineage markers such as Gata4 and Gata6.

Early work showed that imprinted XCI remains in extra-embryonic tissues, including derivatives from PrE[3,4] when Xp reactivation occurs in the pre-Epi[5,6] and was linked to pluripotency factors, such as Nanog and Prdm14[15,16]. The data we present here suggests that X-chromosome reactivation correlates with Epi differentiation. However, reactivation of some genes is not limited to the future Epi cells but initiates independently of lineage segregation in early blastocysts. Indeed, our data suggest that reactivation of some X-linked genes begins before loss of Xist and H3K27me3 and before the strict emergence of PrE and Epi precursor cells, consistent with a previous report[18] (Figs. 2 and 4–6). This suggests that Xp reactivation and the pluripotency program can be uncoupled for some genes such as *Atrx*.

**Fig. 7** UTX is required for proper reactivation of late-reactivated genes. **a** Conditional *Utx* allele: FD = Flp and Dre-recombined conditional allele. Recombination of the third exon of *Utx* by Cre expression gives a knockout FDC allele. **b** Individual *Utx* control and mutant female ICMs analysed by immunolabelling with H3K27me3 (grey), combined with Xist (green) and *Kif4* (red) RNA-FISH in late blastocysts (scale bar 20 μm). Cells below the white line illustrate the cluster of cells that have lost Xist coating and H3K27me3 enrichment on the Xp and are presumably the epiblast. Enlarged nuclei are shown for not reactivated (1, 3) and reactivated cells (2, 4) (scale bar 5 μm). **c** Mean of ICM cells showing enrichment of H3K27me3 on the Xist RNA-coated X-chromosome from E4.5 control (*Utx^{FD/FD}*), heterozygous (*Utx^{FDC/+}*) and mutant (*Utx^{FDC/FDC}*) female blastocysts. Below the graph, the total cell number analysed is indicated, followed by the number of females in brackets. $p$ value <0.0057 (**), <0.0018 (**) and <0.021 (*) between control and heterozygous vs. mutant, respectively, for H3K27me3-Xist-negative cells, H3K27me3-Xist-positive cells and H3K27me3-positive, Xist-negative cells, by two-sided Dunn's test. **d** Example of *Utx* mutant female ICM analysed by immunolabelling with H3K27me3 (grey), combined with Xist RNA (green) at late (E4.5) blastocyst stage. Red and blue arrows pointed cells, respectively, with both Xist and H3K27me3 enrichment and only H3K27me3 enrichment on the Xp (scale bar 10 μm). **e** Percentage (mean) of cells showing biallelic expression for X-linked genes in ICM of E4.5 control (*Utx^{FD/FD}*) and *Utx* mutant (*Utx^{FDC/FDC}*) embryos. *Kif4*, *Rnf12* and *Pdha1* are late- or very late-reactivated genes, when *Abcb7* and *Atrx* are early-reactivated genes (Supplementary Data 2). **f** Control (*Utx^{FD/FD}*) and mutant (*Utx^{FDC/FDC}*) E4.5 blastocysts analysed by immunofluorescence against NANOG (green) and GATA6 (red). DAPI is in dark blue (scale bar 20 μm). Percentage of positive cells for Nanog, Gata6 or both have been summarized as the mean in the graph and total cell number analysed is indicated below, followed by the number of females in brackets. Non-significant (n.s.) by Kruskal–Wallis test. *FDC* Flp, Dre and Cre-recombined knockout allele, *MW* Mann–Whitney non-parametric test

Importantly, some of the early-reactivated genes then show Xp silencing again in E4.0 PrE. This implies a fluctuation in Xi status between E3.5 and E4.0, rather than a constant maintenance of Xp silencing in the future PrE (Fig. 6). Overall, our study highlights the distinct types of behaviour for different X-linked genes when it comes to X-chromosome reactivation. In the case of late-reactivated genes, reactivation is lineage-specific and restrained to the pre-Epi of the mid blastocyst onwards. Later, gene reactivation shows a strong correlation with the presence of NANOG protein (Fig. 2d) and with loss of Xist expression (Fig. 5a) and H3K27me3 enrichment (Fig. 2a). Moreover, loss of Xist RNA coating is the most predictive factor for biallelic expression of the late-reactivated genes (Supplementary Fig. 4c, d).

Our discovery that there are at least two different categories of X-linked genes in terms of reactivation behaviour is important to better understand X-chromosome reactivation and epigenetic reprogramming in general. Interestingly, level of expression and genomic localization are not obvious predictors of X-linked gene behaviour (Supplementary Fig. 2b, d). Our correlative analyses suggest that the dynamic presence of the Myc family might play a role in facilitating some early-reactivated genes to become re-expressed in ICM cells, and revert to a silenced state in PrE cells (Supplementary Fig. 3). In the search for other TFs potentially involved in early reactivation, we used algorithms for motif discovery (see Methods section). Motif comparison analysis of any over-represented motifs in escapees and early-reactivated genes also revealed a correspondence with the TF YY1 (Ying Yang 1), ($p$ value = 0.0002). This motif occurs 2.5 times more frequently in the group of escapees and early-reactivated genes ($n$ = 20, 57 promoters), than in the group of late and very late genes ($n$ = 7, 49 promoters). YY1 is associated with escapees in human and has previously been described to be co-bound to the same binding sites as MYC in mouse ESCs[43,44]. The precise roles of MYC and YY1 proteins in relation to Xp gene activity merits future exploration.

To better understand the degree to which epigenetic chromatin states might be involved in maintaining inactivity, we studied allele-specific H3K27me3 and H3K4me3 enrichment (Fig. 5b). Distinct patterns of differential enrichment of these histone marks was found for early-reactivated genes and escapees (high K3K4me3 on the Xp) and later-reactivated genes (high H3K27me3 on the Xp). These different epigenetic signatures might underlie the distinct transcriptional behaviours of those genes during Xp reactivation. One hypothesis could be that PRC2 is not recruited to early-reactivated genes, avoiding H3K27me3 enrichment at these loci, which could enable a quick response to TFs such as MYC family and/or YY1 in the early ICM.

On the other hand, presence of H3K27me3 at some loci on the Xp may represent a repressive memory mark that maintains silencing at least in the later-reactivated genes. We also found that erasure of H3K27me3 during X-chromosome reactivation is at least partly an active process, as it is delayed in the absence of the H3K27 demethylase, UTX (Fig. 7). The presence of some ICM cells with complete H3K27me3 erasure in *Utx* knockout could be explained by compensation by other demethylases, such as JMJD3 and/or by passive loss of the repressive mark during cell division, although very few cell divisions occur between E3.5 and E4.5 in ICMs[45]. The interference with the kinetics of H3K27me3 loss on the Xp in *Utx* mutants correlates with a decrease in efficiency of reactivation for late-reactivated genes such as *Rnf12* and *Kif4*, but not for early-reactivated genes such as *Atrx* and *Abcb7*. This provides in vivo evidence that Utx may be involved in facilitating the Xp-reactivation process and important insight into the possible mechanisms involved in X-chromosome reactivation and epigenomic reprogramming in general.

In conclusion, our in vivo analysis of the process of Xp reactivation in the ICM reveals that different genes are reactivated by different mechanisms during ICM differentiation. Epigenetic memory of the silenced state involves H3K27me3 maintenance for some X-linked genes but not all. Furthermore, how and why certain genes appear to be excluded from H3K27me3 enrichment on the inactive X chromosome during XCI, and may thus be more prone to rapid reactivation, remains unknown.

Interestingly, expression of several epigenetic modifiers appeared to correlate with X-chromosome reactivation (Supplementary Data 3 and Supplementary Fig. 2g) such as Kdm3a, Kdm3b and Kdm3c (Jumonji C domain-containing protein that demethylates for H3K9 methylation), but also Kdm2b (H3K36-specific demethylase). MacroH2A is enriched on the inactive X chromosome[11] and its variants (H2afy and H2afy2) are expressed in ICM cells (scRNAseq data, expression table in GEO80810). Thus MacroH2A might also repress X-linked gene reactivation, in a redundant fashion with H3K27me3 marks, or specifically for some genes[46]. Future work will be required to determine whether reactivation of the Xp in the ICM also requires erasure or modification of other chromatin states such as H3K9me2 or MacroH2A. Our findings open up the way for a better understanding of the in vivo requirements for epigenetic reprogramming in general.

## Methods

**Mouse crosses and collection of embryos.** All experimental designs and procedures were in agreement with the guidelines from French and German legislations and institutional policies (French ethical committee of animal experimentation: Institut Curie #118 and agreement C75-05-17 for the animal facility, licence number DD24-5131/339/28 for phenotypic analysis of the *Utx* conditional line in the CRTD/BIOTEC animal house).

Mice were exposed to light daily between 7.00 a.m. and 7.00 p.m. Noon on the day of the plug is considered as E0.5. For Figs. 1 and 2, embryos were obtained by natural matings between B6D2F1 (derived from C57BL/6J and DBA2 crosses) females (5–10 weeks old) and males. For the scRNAseq experiments, hybrid embryos were derived from natural matings between C57BL/6J females (5–10 weeks old) crossed with CAST/EiJ males.

To study the absence of Utx in early embryos, females mice carrying heterozygous or homozygous conditional *Utx* alleles (Utx[FD], described in Thieme et al.[30]) and a Cre-driven by *GDF-9* promoter (GDF9-iCre, described in Lan et al.[47]) have been crossed with Utx[FDC/Y] males (Utx[-/Y]). *Utx* control female embryos (Utx[FDC/wt] and Utx[FD/FD]) have been obtained either from the same litters as mutants (from Utx[FD/wt], *GDF-9iCre* females) or after matings between Utx[FD/FD] females with Utx[FD/Y] males.

All embryos were collected between pre-implantation and peri-implantation stages (E3.25–E4.5). Embryos have been classified into early (E3.25–E3.5), mid (E3.75–E4.0) and late (E4.25–E4.5) blastocyst accordingly to morphology, timing and number of cells per ICM (respectively $n$ = 10–25, $n$ = 20–40 and $n$ = 30–55 cells per ICM).

**Immunosurgery for isolation of the ICM.** Pre-implantation blastocyst embryos at stages up to E3.5 (E4.0 for hybrid embryos) were recovered by flushing the uterus with M2 medium (Sigma). Embryos at E3.75 (E4.25 for hybrid embryos) and later were dissected out from the uterus. The embryos were staged on the basis of their morphology and number of cells per ICM.

When applicable, the zona pellucida was removed using acidic Tyrode's solution (Sigma), and embryos were washed twice with M2 medium (Sigma). ICM was then isolated from all stage blastocysts by immunosurgery. Briefly, blastocysts without zona pellucida were quickly cultured in anti-mouse red blood cell serum from rabbit (Rockland) for 30 min then in guinea pig complement serum (Sigma) for 15–30 min. Only TE cells were killed and debris carefully removed with a glass pipette[48].

**RNA-fluorescent in situ hybridization.** RNA-FISH on blastocysts was performed using the exon and intron-spanning plasmid probe p510 for *Xist* (and its antisense *Tsix*) and BAC/Fosmid probes for X-linked genes as described in Supplementary Table 1. Probes were labelled by nick translation with SpectrumGreen, SpectrumRed or SpectrumFRed (Abbott Molecular) using manufacturer's conditions. Before hybridization, probes were precipitated with Cot1 (except for Xist p510 probe), denatured, resuspended in formaldehyde and mixed with 2× hybridization buffer. Fixed embryos were hybridized overnight at 37 °C with labelled probes then washed several times in 2× SSC[9]. Images were acquired using inverted laser scanning confocal microscope with spectral detection (LSM700—Zeiss) equipped

with a 260 nm laser (RappOpto), with a 60× objective and 0.2 μm Z-sections or a 200 M Axiovert fluorescence microscope (Zeiss) equipped with an ApoTome was used to generate three-dimensional (3D) optical sections. Sequential z-axis images were collected in 0.3 μm steps. ICM obtained from $Utx^{FDC/wt}$ females have been PCR-genotyped after image acquisition (details available upon request).

**Immunofluorescence staining**. Briefly, after zona pellucida removal, embryos were fixed in 4% PFA for 15 min and permeabilized in 0.2% sucrose, 0.04% Triton X-100 and 0.3% Tween-20 for 15 min at 37 °C. Embryos were then washed in PBStp (0.05% Triton X-100; 1 mg/ml polyvinyl pyrrolidone (PVP; Sigma)) and blocked in 3% FCS before the primary antibody incubation (overnight at 4 °C). The next day, embryos were washed in PBStp and then incubated for 2 h at room temperature (RT) with corresponding secondary antibodies. After washing, embryos were mounted on coverslips with Vectashield containing DAPI (Vector Laboratories, Burlingame, CA)[49]. All the antibodies used in this study are listed in Supplementary Table 1 along with the information on dilution ratios. Images were acquired using inverted laser scanning confocal microscope with spectral detection (LSM700—Zeiss) equipped with a 260 nm laser (RappOpto), with a 60× objective and 0.2 μm Z-sections. Maximum projections were performed with Image J software (Fiji, NIH).

**Immunofluorescence combined with RNA-FISH**. After zona pellucida removal, embryos were fixed in 3% PFA for 10 min on coverslips then permeabilized (Triton/VRC/PBS) on ice for 15 min. Embryos were then blocked in PBS/BSA for 15 min at RT, incubated with primary antibodies with RNAase inhibitor (1–2 h at RT) and then secondary antibodies (30 min at RT)[5]. Embryos were then again fixed in 4% PFA for 10 min at RT. RNA-FISH was then performed as mentioned previously. Images were acquired using inverted laser scanning confocal microscope with spectral detection (LSM700—Zeiss) equipped with a 260 nm laser (RappOpto), with a ×60 objective and 0.2 μm Z-sections or a confocal wide-field Deltavision core microscope (Applied Precision—GE Healthcase) with a ×60 objective (1.42 oil PL APO N) and 0.2 μm Z-sections or a 200 M Axiovert fluorescence microscope (Zeiss) equipped with an ApoTome was used to generate 3D optical sections. Sequential z-axis images were collected in 0.3 μm steps. Images were analysed using ImageJ software (Fiji, NIH).

ICMs obtained from $Utx^{FDC/wt}$ females were PCR-genotyped after image acquisition (details available upon request).

All the antibodies and probes used in this study are listed in Supplementary Table 1 along with the information on dilution ratios.

**Single-cell dissociation from ICMs**. To isolate individual cells, we incubated the ICM in TrypLE solution for 5 min (Invitrogen). After incubation, each blastomere was mechanically dissociated by mouth pipetting with a thin glass capillary. Single cells were then washed three times in PBS/acetylated BSA (Sigma) before being manually picked into PCR tubes with a minimum amount of liquid. We either directly prepared the complementary DNA amplification or kept the single cells at −80 °C for future preparation.

**Single-cell RNA amplification**. PolyA+ mRNA extracted from each single cell was reverse-transcribed from the 3′UTR and amplified following the protocol by Tang et al.[36] Care was taken to process only embryos and single blastomeres of the highest quality based on morphology, number of cells and on amplification yield (Supplementary Table 1). Additional RT-specific primer for Xist amplification have been added in the lysis buffer, which contains 100 nM universal RT-primer UP1 and 15 nM Xist-specific RT primer ES323 (ATATGGATCCGGCGCGCCGTCGA C(T)24 GCAAGGAAGACAGACACACAAAGCA).

Published scRNAseq samples of E3.5 TE and ICM from the same interspecific cross and the reverse cross and amplified following the same method have been added to our analysis (GSE80810; Borensztein et al.[7]).

**Single-cell libraries and deep sequencing**. After single-cell amplification, each single-cell gender has been analysed by quantitative PCR for Xist and Y-linked genes Eif2s3y, Uty and Ddx3y. Single-cell libraries were prepared from 34 females samples, which have passed quality controls according to the manufacturer's protocol (Illumina) and were deeply sequenced on an Illumina HiSeq 2500 instruments in single-end 50-bp reads (Supplementary Data 1).

**Quality control and filtering of raw data**. Quality control was applied on raw data following the method by Borensztein et al.[7] Briefly, sequencing reads characterized by at least one of the following criteria were discarded from the analysis:

1. >50% of low-quality bases (Phred score <5).
2. >5% of N bases.
3. At least 80% of AT rate.
4. >30% (15 bases) of continuous A and/or T.

**Estimation of gene expression levels**. RNA reverse transcription allowed sequencing only up to an average of 3 kb from the 3′ UTR. To estimate transcript

abundance, read counts were thus normalized on the basis of the amplification size of each transcript (retrotranscribed length per million mapped reads, RPRT) rather than on the basis of the size of each gene (RPKM), based on the method by Borensztein et al.[7] To avoid noise due to single-cell RNAseq amplification technique, only well-expressed genes (RPRT >4) were considered in our allele-specific study. A threshold of RPRT >1 was applied to consider a gene as expressed (Figs. 3, 4 and Supplementary Figs. 3a and 4a). Low-expressed genes were excluded from the analysis in order to avoid amplification biases due to single-cell PCR amplification.

**Allele-specific RNA-seq pipeline**. Allele-specific RNA-seq analysis pipeline were adapted from the method by Borensztein et al.[7] and applied to our data, using the same parameters, parental genomes, annotations and SNPs files. Briefly, we have filtered the single-nucleotide polymorphisms (SNPs) on their quality values (F1 values), thanks to SNPsplit tool (v0.3.0)[50]. A SNP on chr:X 37,805,131 (mm10) in Rhox5 gene, annotated A for C57BL/6J and G for all other strains (included C57BL/6NJ) was discarded because it was missing in our samples. After reconstruction of both maternal (C57BL/6J) and paternal (CAST/EiJ) genome, allele-specific read alignment was performed with TopHat2 (v2.1.0)[51] software. The SAMtools mpileup utility (v1.1)[52] was then used to extract base-pair information at each genomic position. At each SNP position, the numbers of paternal and maternal alleles were counted. The threshold used to call a gene informative was five reads mapped per single SNP, with a minimum of eight reads mapped on SNPs per gene, to minimize disparity with low-polymorphic genes. The allele-specific origin of the transcripts (or allelic ratio) was measured as the total number of reads mapped on the paternal genome divided by the total number of paternal and maternal reads for each gene: allelic ratio = paternal reads/(paternal + maternal) reads.

Genes were thus classified into two categories:

1. Monoallelically expressed genes: allelic ratio value ≤0.15 or ≥0.85.
2. Biallelically expressed genes: allelic ratio value >0.15 or <0.85.

**PCA and hierarchical clustering**. Gene count tables were generated using HTSeq software (v0.6.1). Rlog function from DESeq2 R-package (v1.12.2) was used to normalize the raw counts data, with filter thresholds as described[7]. To identify the cell origin of our samples, PCA and hierarchical clustering (Pearson correlation—Ward method) on normalized data of 23 lineage-specific factors (Fig. 3) were performed using plotPCA function from DESeq2 R-package and hclust function implemented in the gplots R-package (v3.0.1), respectively.

**Heatmap of the X-chromosome**. Data from informative genes were analysed if the gene was expressed (RPRT >4) in at least 25% of the single cells (with a minimum of two cells except for TE) in a particular developmental stage. To follow reactivation, we decided to focus on genes at least expressed in both PrE and Epi lineages at E4.0 stage. Mean of the allelic ratio of each gene is represented for the different stages. Same list of genes was used for all heatmaps (116 genes). Only single cell from the same interspecific cross have been used (C57BL/6J females × CAST/EiJ males, mentioned as BC cross below) as different genes could follow different kinetics in a strain-specific manner[7].

**Definition of the timing of reactivation**. A minimum of 20% of expression from the Xp has been used as a threshold to call a gene as reactivated in the female samples. Adapting the method used in Borensztein et al.[7], we have automatically associated X-linked genes that become biallelic in the ICM at E3.5 (allelic ratio ≤0.15 in TE or inactivated at the same stage in Borensztein et al.[7] and >0.20 in ICM at E3.5) stage to early-reactivated gene class and in the Epi at E4.0 stage to late-reactivated gene class (allelic ratio equals NA or in TE, NA or ≤0.20 in ICM at E3.5 and >0.20 in Epi at E4.0). X-linked genes showing very late reactivation (0.15 ≤ allelic ratio in TE at E3.5 and 0.2 ≤ allelic ratio in other stages) in all stages are categorized as not yet reactivated genes. Finally, the last group represents genes that are escaping imprinted Xp inactivation (allelic ratio >0.15 in all stages, or NA at E3.5 and allelic ratio >0.15 in the other stages). Some genes could not be associated to a gene class due to several missing values in the decisive stages, however, classes have been associated to them if RNA-FISH data was available or in case of imprinted genes (e.g., Xlr3a and Xist classed as 'others').

**Correlation between autosomal and X-linked gene expression**. Correlation and anti-correlation between gene expression levels (autosomes and X chromosomes) and percentage of X-linked gene reactivation (allelic ratio >0.2 for X-linked genes) was measured by Pearson correlation and Benjamini–Hochberg correction and are provided in Supplementary Table 3. BC and CB (reverse cross) (only for E3.5 TE) female single cells have been used in this analysis.

Gene ontology has been made for the top correlated genes (q value <0.05) with the Gene Ontology Project[53] and AMigo software[54].

**Allele-specific H3K27me3 and H3K4me3 ChIPseq analysis**. H3K27me3 and H3K4me3 enrichments in ICM were taken, respectively, from Zheng et al.[38] and

Zhang et al.[39] Bed files of either maternal or paternal chromosomes for both marks were used to assess the enrichment of either marks at 5 kb around their TSS. For genes having several TSS, position of start (for gene on the +strand) or end (for genes on the −strand) of the gene were taken. Score for each 100-pb window containing enriched marks were summed (by Custom R scripts (R Core Team (2015). R: a language and environment for statistical computing. R Foundation for Statistical Computing, Vienna, Austria https://www.R-project.org/)). For genes whose length was below 5 kb, gene size was taken as window. Distribution of gene size for each group was not significantly different.

**TF-binding sites analysis**. Nanog, Oct4, Sox2, Myc, Mycn, Klf4, Esrrb and Tcfcp2l1 binding sites and their TF-gene-associated score for each gene were taken from ChIPseq experiments in mouse ESCs, previously published in Chen et al.[40] The TF-gene-association score was calculated by Chen et al.[40], between each pair of gene and TF, from 0 to 1, with a higher score linked to a higher probability of the gene to be a direct target of the TF. For pluripotency factor analysis (Supplementary Fig. 3b), Nanog, Oct4, Sox2, Klf4, Esrrb and Tcfcp2l1 scores have been summed for each X-linked gene of the different reactivation-timing groups (Supplementary Data 2). In Supplementary Fig. 3c, Myc and Mycn scores have been summed. Mycl sites were not analysed in Chen et al[40].

**Motif discovery analysis**. RSAT oligo-analysis[55] was used to search for over-represented motifs in promoters (−700/+299 nts relative to TSS) of X-linked genes in escapees, early-, late- and very late-reactivation classes. Since the number of genes per class is too low to obtain high confidence results, we pooled genes by similar behaviour, with escapees and early-reactivated genes in one group and late and very late in another one. One non-repetitive motif was found over-represented in the first group. This motif was compared to a database of known TF motifs using Tomtom (MEME Suite)[56] and only one correspondence was found with $E$ value <1, that of the TF YY1 motif ($p$ value = 0.0002, $E$ value = 0.27, $q$ value = 0.54). FIMO (MEME Suite)[56] was used to determine the occurrences of this motif in each group of genes, and only matches with a $p$ value <0.0001 were considered.

**Statistics section**. Kruskal–Wallis and post hoc test were used to analyse non-parametric and unrelated samples.

The statistical significance has been evaluated through two-sided Dunn's multiple comparison test with Benjamini–Hochberg correction and Kruskal–Wallis analysis of variance. $p$ values are provided in the figures, figure legends and/or main text. Enrichment of histone marks has been evaluated thanks to non-parametric Wilcoxon test.

**Data availability**. The Gene Expression Omnibus (GEO) accession number for the scRNAseq data produced in this paper is GSE89900. Public data sets for scRNAseq and ChIPseq used in the manuscript are, respectively, GSE80810 and GSE76687 (H3K27me3), GSE71434 (H3K4me3), GSE11431 (pluripotency factors). All other data and codes for allelic mapping are available from the corresponding author upon reasonable request.

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

## Acknowledgements

We are grateful to P. Gestraud for help in statistical analysis. We thank the pathogen-free barrier animal facility and the Cell and Tissue Imaging Platform-PICT-IBiSA (member of France–Bioimaging) of Institut Curie. We acknowledge C.A. Penfold and the members of E.H. and A.S. laboratories for feedbacks and critical inputs. This work was funded by fellowships from Région Ile-de-France (DIM STEMPOLE), Fondation Recherche Médicale (FRM SPE20150331826) and a Marie Sklodowska-Curie Individual Fellowship (H2020-MSCA-IF-2015—no. 706144) to M.B., CELLECTCHIP (ANR-14-CE10-0013) to E.H. and M.B., the Paris Alliance of Cancer Research Institutes (PACRI-ANR) to L.S., ERC Advanced Investigator award (ERC-2010-AdG—no. 250367), EU FP7 grants SYBOSS (EU seventh Framework G.A. no. 242129), MODHEP (EU seventh Framework G.A. no. 259743), La Ligue, Fondation de France, Labex DEEP (ANR-11-LBX-0044) part of the IDEX Idex PSL (ANR-10-IDEX-0001-02 PSL) and ABS4NGS (ANR-11-BINF-0001) to E.H., France Genomique National infrastructure (ANR-10-INBS-09) to E.H., N. S. and E.B., a grant-in-aid from MEXT and JST-ERATO to I.O. and M.S., a JSPS KAKENHI Grant (Number JP25291076) to I.O. and a DFG grant (SPP1356) to K.Ana. G.G. is supported by the central grant to the LMB by the MRC (U105178808).

## Author contributions

I.O., M.B. and E.H. conceived the study, with input from M.S. and A.S. I.O., M.B. and K. Ana. performed most of the IF/RNA-FISH experiments. K.Anc. performed the IF experiments. C.P., P.D. and K.A. helped for IF/RNA-FISH experiments and acquisition. M.B. performed single-cell RNA amplification, and C.-J.C. performed the transcriptome library preparation and sequencing. L.S. and M.B. analysed the scRNAseq data, and bioinformatics was supervised by N.S. and E.B. G.G. and R.G. performed, respectively, the ChIPseq and the motif discovery analysis. M.B., I.O. and E.H. wrote the paper with input from all co-authors.
