## [Peer Review File · Nature Communications]

Reviewers' comments:

Reviewer #1 (Remarks to the Author):

This paper presents a detailed analysis of the reversal of XCI that occurs in the inner cell mass before random XCI occurs. The manuscript follows, and uses some of the data from the authors' recent paper examining early imprinted XCI (2 cell to 32 cell stage focus with some overlap of blastocyst examination). There has been considerable variation in descriptions of the events of early inactivation, which are challenging to follow, so such a detailed analysis should be well-received. The authors find considerable variation in timing of reactivation of genes, and are able to demonstrate differences in chromatin marks that may underlie such differences and by assessing Utx deficient blastocysts present support for an active H3K27me3 demethylation process for some X-linked gene reactivation.

The paper includes a lot of acronyms, and care should be taken to ensure they are defined. Line 84 introduces trimethylation of histone H3 Lysine 27, but not the acronym, which is used later. Similarly XCR (used in the results) could be defined on line 100. The introduction is concise, yet complete, although the focus on Utx rather than Jmjd3 should be discussed.

In discussing the concordancy with the Williams et al study it would be helpful to describe the mouse strains assessed, particularly given the recent support for strain-specific effects (including the Borensztein et al, 2017 paper).

The PCA analysis of Supp. Figure 1b (discussed on lines 303-307) suggested that "whatever the timing of XCI (early, mid and late silenced as classified in Borensztein et al., 2017), genes are more prone to be late or very late reactivated in the ICM". This sentence is difficult to follow, but suggests that the early reactivated genes (Figure 3a) may be unique - (as seen for their chromatin marks), but are there other differences in genomic location? The higher H3K4me3 suggests that they may show higher expression levels - is that true?

Comparison with allelic ChIP data. As alluded to in the comment above, the H3K4me3 data on the Maternal X (supplemental) suggest that there may be expression differences between escapees or early reactivating genes compared to late and very late reactivating genes. Therefore are expression levels (or chromatin marks) predisposing to reactivation and escape? It seems that this should be discussed further.

Figure 1. The legend does not describe the white line (nor how it was assigned). Clarification is needed as to how many cells were counted, assuming the n=20 or 10 is the number of blastocysts?

Figure 2. The difference between E3.5 and E3.75 is dramatic for the nanog positive cells, again how many cells were compared? Were the mice from the same breeding period? The methods suggest E3.5 were flushed while E3.75 was removed by immunosurgery? Were the cell counts significantly increased in 3.75 as anticipated given that later stages were staged by cell count, if my interpretation of the methods are correct? Why does E3.75 not require highlighting of the ECM? The description of H3K27me3 as 'white', I presume reflect greyscale representation? It would be helpful to label genes that escape XCI.

Figure 3c. It would be clearer to me if this panel were split into left and right (3d) so that the legend would be clear as to which panel was being assessed. The oblong nature of 3c (left) offset the symmetry.

Figure 4c - It would be helpful if the stage (E3.5-E4.0) for the chromatin data was included in the Figure legend.

Reviewer #2 (Remarks to the Author):

The manuscript entitled "Differential epigenetic memory states and reactivation kinetics of the inactive X chromosome in the inner cell mass" by Borensztein et al. investigates X chromosomereactivation in preimplantation blastocyst stage female mouse embryos using single cell expression methods. The authors show that reactivation of a subset of genes occurs in inner cell mass cells when Xist and histone H3 lys27 methylation is present on the inactive X chromosome. However, the majority of genes appear reactivated after Xist is displaced. Notably, the authors observe rapid resilencing of early reactivators when extraembryonic endoderm expression profiles are established. This finding is interesting and could point towards a dynamic chromosome inactivation phase inherent to the lineage decision process. A second notable finding is a delay in loss of histone H3lys27 methylation in embryos lacking the demethylase UTX. These results suggest that H3 methylation could be contributing to maintaining repression of late reactivators and early reactivating genes would not be affected. Analysis further suggests that genes that early reactivate or escape from silencing are more likely to be bound by Myc transcription factors.

In conclusion this study contains data of very high technical quality and makes observations that are interesting for researchers in X chromosome inactivation. A few points should be addressed:

Major points

1. The authors investigate PcG marks that appear not strongly linked with silencing in X inactivation. However, macroH2A is not considered in the discussion that might have a role in maintaining repression at least temporarily. Could a statement be added if macroH2A histone variants are expressed and localized to the inactive chromosome.
2. Page 13: On the important point of differentiating genes that reactivate early and late the conclusion seems not clear. Is reactivation correlated with some transcriptional property? From the Myc binding maybe promoter strength could be considered. Alternatively, chromatin organization and structure might be considered. Is there any idea which is a more likely mechanism or can one be ruled out?
3. Page 7 line 136: Loss of K27me in clusters of cells is interpreted as loss in one cell cycle could be active. This would be true if all cells cycle with the same rate. However, clustering could indicate that few cells replicate multiple times and loss of methylation could be still passive. Can this scenario be ruled out?
4. Page 14 line 303 - 305: can this sentence be clarified: what are genes that are efficiently silenced? Is silencing not a prerequisite for being able to become reactivated later on?

Minor points:

- a) Abstract line 51: initial enrichment in K27me ... correlates with speed of reactivation. Should this be anticorrelated? or lack of enrichment?
- b) Page 18 last sentence and title, abstract: .. different epigenetic memory states seems odd when the data suggest Myc TF binding as the clearest difference between early and late reactivators. It could be omitted from the title without losing the focus.
- c) Figure 4A: The paternal expression of Xist is expected but of Ogt and YLpf6 is surprising. Can a brief explanation be added.

Reviewer #3 (Remarks to the Author):

Preimplantation mammalian embryos feature a number of interesting global epigenetic processes and the dynamics of the X chromosome inactivation/re-activation is certainly one of them. The manuscript by Borensztein and colleagues provides a nice addition to our current (limited) knowledge about the process of Xp inactivation in the ICM of female mouse embryos. Using RNA FISH and scRNA Seq on cells isolated from E3.5-E4.0 mouse female embryos, the authors show that the reactivation of the Xp is more complicated than previously assumed. Early reactivating genes do not seem to be marked by H3K27me3 and their biallelic expression precedes the loss of Xist expression. To the contrary, late re-activating genes show enrichment of H3K27me3, the biallelic expression of these genes correlates with the loss of H3K27me3 and of Xist coating. Re-activation of these genes is also susceptible to the loss of Utx - H3K27me3 demethylase.

Overall, this is a nice and interesting story that provides a number of interesting insights.

I have a few comments that I think should be addressed:

1) Probably the most surprising finding is that some early re-activating genes appear to be biallelically expressed prior to the lineage segregation in the ICM. As the PrE cells are supposed to keep inactive Xp, the authors suggest that upon lineage segregation, these genes will be specifically re-silenced on the Xp in PrE cells. I wonder whether this is something that can actually be demonstrated? Transient erosion of the Xp could potentially lead to the epigenetic differences between the Xp in TE and PrE cells, is there any evidence for this?

The authors claim that there is considerable transcriptional heterogeneity in the cells of the early ICM. However, looking at the PCA analysis, a possible interpretation might be that there is some separation already between the cells that might become future epiblast vs PrE. Can this interpretation be excluded?

Also, if this was the case, can it be excluded that the biallelic expression of the early reactivating genes can be seen only in the cells of early ICM that will eventually contribute only to the epiblast? (ie there is no transient biallelic expression in cells that will become PrE)?

2) Correlation is made between transcriptional reactivation of the early reactivating genes and myc (in reference to the reported hypetranscription in the ICM). I was wondering whether the authors performed an unbiased analysis of the TF binding site enrichment in the promoter regions of the early re-activating genes (vs late re-activating genes)? Why was myc specifically selected?

3) What is the expression level of Utx in cells of early embryos (scRNASeq data)? Is there any difference between the cells of individual lineages? Are there any other candidate demethylases that could compensate for the loss of Utx?

4) Fig3: a) and b) there should be a distinct colour label for TE cells; there seem to be more early E3.5 data points in the PCA analyses than in the hierarchical clustering (c) and d). What is the reason for this?

Reviewer #4 (Remarks to the Author):

In this manuscript, Borensztein and colleagues performed a set of experiments to decipher the X-chromosome reactivation dynamics in the mouse embryonic development. Using a combination of

RNA FISH and single-cell sequencing techniques, the authors discover that bi-allelic gene re-activation in the inner cell mass display different temporal dynamics that are, in part, linked with H3K27me3 epigenetic memory. The findings are topical and address factors that contribute to regulating the X-chromosome during development. In general, this study is well-designed and organized in a logical manner. However, several issues dampen my enthusiasm for the paper.

Major comments:

- 1) As shown in Figure 2, there are eight X-linked genes that show differential reactivation timing dynamics. The differential reactivation is not correlated with the linear distance from the Xist locus. Previous studies have shown that the spread of Xist RNA and gene repression is associated with the 3-dimensional architecture of the X-chromosome. The authors should cross-reference to such Xist-coating maps to determine any relationship between reactivation dynamics and Xist coating/H3K27me3 localization.
- 2) Related to the analysis of Single-cell RNA-sequencing, the authors should provide a table indicating the number of X-linked genes covered and the total number of SNPs that were used for each cell. In addition, the single cell RNAseq protocol as described in this paper is well known to have 3' bias that could affect the SNP analysis. The authors should try an independent single-cell amplification approach, such as SMRT-seq2, which is capable of full length transcript profiling. The number of profiled TE cells (n=3) is also too few for proper statistical comparison given the large variation riddled in single-cell RNA-sequencing datasets. A subset of samples from male cells as a negative control would also be appropriate.
- 3) As shown in Figure 3, the mouse single-cell RNA-seq data have previously been performed by Deng and Sandberg et al, the authors should use this orthogonal piece of data for cross-validating their findings.
- 4) In the analysis of allele-specific expression, the authors cite their paper (now published in NSMB) that uses an analytical pipeline in which 15% is used as a cutoff for bi-allelic vs mono-allelic expression. Is this cut-off arbitrary, why not 10% or 5%?
- 5) As shown in Figure 4, are there any transcriptome wide differences between sister cells from the same lineage that differ in % bi-allelic gene expression? In other words, does X-reactivation correlate with any other autosomal changes?
- 6) To examine the role of H3K27me3 in X-linked gene reactivation, the authors used the Utx mutant embryos and see some effects. It would be informative if single-cell sequencing data was available for comparison with Figure 4. Finally, for the data analysis in Figure 5, the authors should correct for Type I error in the KW test as some of the p-values may become marginal.

Point by Point Rebuttal for Borensztein, Okamoto et al

We would like to thank the reviewers for their constructive comments and helpful suggestions. We have been able to address most of the points raised in our revised manuscript. We feel that these changes have greatly improved our paper. Below we provide a point-by-point rebuttal to all of the comments made by the reviewers.

Reviewer 1:

We thank the reviewer 1 for his/her helpful suggestions and comments, all of which we answer below.

This paper presents a detailed analysis of the reversal of XCI that occurs in the inner cell mass before random XCI occurs. The manuscript follows, and uses some of the data from the authors' recent paper examining early imprinted XCI (2 cell to 32 cell stage focus with some overlap of blastocyst examination). There has been considerable variation in descriptions of the events of early inactivation, which are challenging to follow, so such a detailed analysis should be well-received. The authors find considerable variation in timing of reactivation of genes, and are able to demonstrate differences in chromatin marks that may underlie such differences and by assessing Utx deficient blastocysts present support for an active H3K27me3 demethylation process for some X-linked gene reactivation.

1-The paper includes a lot of acronyms, and care should be taken to ensure they are defined. Line 84 introduces trimethylation of histone H3 Lysine 27, but not the acronym, which is used later. Similarly XCR (used in the results) could be defined on line 100.

We would like to thank the reviewer for this comment. Care has now been taken to define each acronym used in the text and XCR is no longer used to simplify comprehension.

2-The introduction is concise, yet complete, although the focus on Utx rather than Jmjd3 should be discussed.

We agree with the reviewer on the lack of a discussion on H3K27m3 demethylases. We have now introduced the other H3K27 demethylases in the introduction, with a discussion on our focus for UTX rather than JMJD3.

Introduction, Pages 5-6, lines 110-119:

“Genome-wide removal of the tri-methylation of H3K27 may be catalysed by the JmjC-domain demethylase proteins: UTX (encoded by the X-linked gene Kdm6a), UTY (a Y-linked gene) and JMJD3 (encoded by Kdm6b), (Hajkova et al. 2008; Hong et al. 2007; Agger et al. 2007; Lan et al. 2007). Diverse roles have been proposed for these demethylases (Shpargel et al. 2012; Mansour et al. 2012; Yang et al. 2016). JMJD3 appears to inhibit reprogramming (Zhao et al. 2013), whereas UTX plays a role in differentiation of the ectoderm and mesoderm (Morales Torres, Laugesen, and Helin 2013) and has been proposed to promote somatic and germ cell epigenetic reprogramming (Mansour et al. 2012). Interestingly, the Utx gene escapes from X-chromosome inactivation (ie is transcribed from both the active and inactive X chromosomes) (Greenfield et al. 1998). This raises the intriguing possibility that Utx might have a female-specific role in reprogramming the Xi in the inner cell mass of the mouse blastocyst.”

3-In discussing the concordancy with the Williams et al study it would be helpful to describe the mouse strains assessed, particularly given the recent support for strain-specific effects (including the Borensztein et al, 2017 paper).

We agree that strain-specific effects, as well as the timing of embryo collection, could explain some of the discrepancies with the Williams *et al* study for specific genes. Indeed, all our wild-type IF and/or RNA-FISH analysis was performed on B6D2F1 mice (derived from C57BL/6J and DBA2 crosses) and allele-specific scRNAseq on C57BL/6J and Castaneus hybrids, whereas the other study used CD-1 mice for RNA-FISH and hybrids CD-1 and molossinus for allele-specific RT-PCR. Moreover, the timing of the blastocyst analyses may also have been slightly different between our two studies. In our manuscript, embryos are classified into early (E3.25-E3.5), mid (E3.75-E4.0) and late (E4.25-E4.5) blastocyst accordingly to morphology and number of cells per ICM (respectively n=10-25, n=20-40 and n=30-55 cells per ICM). In the Williams *et al* study blastocysts were classified only based on timing of collection, ie early (E2.75-E3.0) and mid (E3.5-E3.75) respectively.

We have now modified our manuscript accordingly.

Results, Page 8, lines 165-169:

“These results are only partially concordant with the Williams et al. study (Williams et al. 2011), however the apparent discrepancy in Rnf12 reactivation timing may be due to differences in the

exact stages of blastocyst development examined, or to the different mouse strains used (B6D2F1 and B6xCast here, compared to CD-1 and CD-1xJF1 in Williams et al)."

4-The PCA analysis of Supp. Figure 1b (discussed on lines 303-307) suggested that "whatever the timing of XCI (early, mid and late silenced as classified in Borensztein et al., 2017), genes are more prone to be late or very late reactivated in the ICM". This sentence is difficult to follow, but suggests that the early reactivated genes (Figure 3a) may be unique – (as seen for their chromatin marks), but are there other differences in genomic location? The higher H3K4me3 suggests that they may show higher expression levels – is that true?

We agree with the reviewer that our sentence was confusing. We intended to highlight the absence of mirroring kinetics between silencing and reactivation kinetics.

To answer the reviewers' question about other specific features of the early reactivated genes, we analysed their position along the X chromosome, their location inside or in close proximity to first *Xist* entry sites, as well as their level of expression (Supplementary Figure 2). No significant association between genomic localisation and reactivation timing has been found even if early reactivated genes have a tendency to be further from the *Xist* locus compared to late and very late reactivated genes (Supplementary 2b). We have now changed the text accordingly.

Results, Pages 14-15, lines 318-329:

*"Correspondence analysis revealed that kinetics of reactivation of X-linked genes does not mirror their kinetics of silencing (Supplementary Figure 2a). Clearly the timing of reactivation is not simply about the lapse of time since silencing was initiated, nor about the location of a gene along the X chromosome (Supplementary Figure 2b). Although a slight tendency was observed for late and very late reactivated genes to be in close proximity of the *Xist* locus, *Atrx* and *Abcb7* genes are both silenced early, lie close to the *Xist* genomic locus and yet are also reactivated early (Borensztein et al. 2017; Patrat et al. 2009; Deng et al. 2014). Furthermore, our previous work revealed that although early silenced genes preferentially lie inside the first *Xist* "entry" sites as defined by Engreitz et al in ESCs, the late and very late reactivated genes failed to show any significant proportion correlation with *Xist* entry sites (Supplementary Figure 2c) (Borensztein et al. 2017; Engreitz et al. 2013). Gene expression level was also not found as an obvious predictor of early or late reactivation (Supplementary Figure 2d)."*

5-Comparison with allelic ChIP data. As alluded to in the comment above, the H3K4me3 data on the Maternal X (supplemental) suggest that there may be expression differences between escapees or early reactivating genes compared to late and very late reactivating genes. Therefore are expression levels (or chromatin marks) predisposing to reactivation and escape? It seems that this should be discussed further.

As suggested by the reviewer, higher H3K4me3 enrichment suggests higher expression. We have now compared the level of expression of X-linked genes depending of their low, mid or high enrichment in H3K4me3 marks and show a significant increase in expression for the highly enriched genes compared to the low one on both paternal and maternal chromosomes (Supplementary Figure 2f). However, when plotting the level of expression of X-linked genes as a function of their reactivation timing class, and independently of their enrichment in H3K4me3, no difference could be found (Supplementary Figure 2d). We therefore conclude that the level of expression is not a key feature of early-reactivated genes, suggesting that paternal enrichment of H3K4me3 could be a consequence of biallelic expression of those genes.

See quote for question 4-Results (Pages 14-15).

Results, Pages 15-16, lines 344-348:

“As expected, we confirmed that H3K4me3-highly enriched genes are globally more highly expressed than lowly enriched genes (Supplementary Figure 2f). However as no association was found between a high level of expression and early reactivation, we hypothesize that paternal enrichment of H3K4me3 could be a consequence of biallelic expression of early reactivated genes.”

6-Figure 1. The legend does not describe the white line (nor how it was assigned). Clarification is needed as to how many cells were counted, assuming the n=20 or 10 is the number of blastocysts?

The legend was indeed lacking important information and we have modified it and the related figures accordingly. Number of female blastocysts and total number of analysed cells are provided under each graph for Figures 1, 2 and 5.

Figure Legends for Figures 1, 2 and 5.

“The cells below the white line illustrate the cluster of cells that have lost Xist RNA coating and H3K27me3 enrichment on the Xp and are presumably the epiblast.”

“Below the graph the total cell number analysed is indicated, followed by the total number of female embryos analysed in brackets.”

7-Figure 2. The difference between E3.5 and E3.75 is dramatic for the nanog positive cells, again how many cells were compared?

Cell numbers have been added to the figures now. In this particular experiment, E3.75 embryos were classified as mid blastocysts according to their ICM cell number and their morphology.

8-Were the mice from the same breeding period?

A statement has been added in Methods. All our wild-type embryo IF and/or RNA-FISH experiments were performed on the same strain (B6D2F1), using 5-10 weeks old females.

Methods, Page 31, lines 682-684:

“For Figures 1 and 2, embryos were obtained by natural matings between B6D2F1 (derived from C57BL/6J and DBA2 crosses) females (5-10 weeks old) and males.”

9-The methods suggest E3.5 were flushed while E3.75 was removed by immunosurgery?

Indeed, E3.5 and E3.75 stage embryos were collected differently. Early blastocysts (E3.25-E3.5) are not implanted in the uterus and still have their zona pellucida. Thus we can retrieve them by simply flushing them out of the uterus. By the mid-blastocyst stage (E3.75-E4.0) however, the embryos have lost their zona pellucida and dissection of the uterus allows us to collect the embryos without damaging them by flushing. Once the blastocysts from all stages have been collected, immunosurgery is performed, to remove the trophectoderm cells and expose the inner cell mass cells for analysis.

The Methods section has been modified accordingly.

Methods, Pages 31-32, lines 698-704:

“Pre-implantation blastocyst embryos at stages up to E3.5 (E4.0 for hybrid embryos) were recovered by flushing the uterus with M2 medium (Sigma). Embryos at E3.75 and later were dissected out from the uterus. The embryos were staged on the basis of their morphology and number of cells per ICM.

When applicable, the zona pellucida was removed using acid Tyrode’s solution (Sigma), and embryos were washed twice with M2 medium (Sigma). Inner Cell Mass (ICM) was then isolated

from all stage blastocysts by immunosurgery as previously described (Matsui, Goto, and Takagi 2001)”

10-Were the cell counts significantly increased in 3.75 as anticipated given that later stages were staged by cell count, if my interpretation of the methods are correct?

Indeed the reviewer has understood correctly: blastocysts were staged by cell number. Mid blastocysts, (20-40 cells/ICM) were collected between E3.75 and E4.0. In Figure 2d-f, a significant increase in the number of cells per ICM between E3.5 and E3.75 can be seen, with a mean of 21 and 33 cells/ICM respectively.

Care has been taken to clarify the Methods and timing of collection for early, mid and late blastocysts. Interestingly, mid blastocysts from the hybrid cross (C57BL/6J x Castaneus), used for allele-specific scRNAseq, were never seen around E3.75 and always had to be collected at E4.0.

Methods, Page 31, lines 692-695:

“All embryos were harvested between pre-implantation to peri-implantation stages, respectively between E3.25 to E4.5. Embryos have been classified into early (E3.25-E3.5), mid (E3.75-E4.0) and late (E4.25-E4.5) blastocyst accordingly to morphology, timing and number of cells per ICM (respectively n=10-25, n=20-40 and n=30-55 cells per ICM).”

11-Why does E3.75 not require highlighting of the ECM?

We are not sure what the question is – but we believe that the reviewer is asking why the cells of the ICM are not sub-divided into two sub-population to highlight the cells, which have lost Xist enrichment and are NANOG-positive, as was done in Figure 1.

In Figure 2, the orientation of the E3.75 inner cell mass we have chosen for illustration makes it difficult to draw a white line in between potential epiblast and PrE. We therefore decided to highlight just the NANOG-positive cells, instead.

12_The description of H3K27me3 as ‘white’, I presume reflect greyscale representation?

Legends have been modified accordingly. We thank the reviewer for pointing out this error.

13-It would be helpful to label genes that escape XCI.

A sentence has been included in the Figure 2b legend to label *Atp6ap2* gene as an escapee.

Figure Legend for Figures 2b

“*Atp6ap2* gene is known to escape XCI in 60% to 80% of blastocyst cells (Patrat et al. 2009) and used as a control of the experiment.”

14-Figure 3c. It would be clearer to me if this panel were split into left and right (3d) so that the legend would be clear as to which panel was being assessed. The oblong nature of 3c (left) offset the symmetry.

We agree with the reviewer that the split between our two panels greatly improves the comprehension of our Figure 3 legend. We have modified Figure 3 and the legend accordingly.

15-Figure 4c – It would be helpful if the stage (E3.5-E4.0) for the chromatin data was included in the Figure legend.

As requested, a short sentence about the chromatin data has been included to the Figure legend 4c.

Figure Legend for Figure 4c

“Low cell ChIPseq have been performed with ICM cells of pre-implantation embryos (pooled between E3.5-E4.0) after immunosurgery of the ICM (Zheng et al. 2016)”.

Reviewer 2:

The manuscript entitled "Differential epigenetic memory states and reactivation kinetics of the inactive X chromosome in the inner cell mass" by Borensztein et al. investigates X chromosomereactivation in preimplantation blastocyst stage female mouse embryos using single cell expression methods. The authors show that reactivation of a subset of genes occurs in inner cell mass cells when Xist and histone H3 lys27 methylation is present on the inactive X chromosome. However, the majority of genes appear reactivated after Xist is displaced. Notably, the authors observe rapid resilencing of early reactivators when extraembryonic endoderm expression profiles are established. This finding is interesting and could point towards a dynamic chromosome inactivation phase inherent to the lineage decision process. A second notable finding is a delay in loss of histone H3lys27 methylation in embryos lacking the demethylase UTX. These results suggest that H3 methylation could be contributing to maintaining repression of late reactivators and early reactivating genes would not be affected. Analysis further suggests that genes that early reactivate or escape from silencing are more likely to be bound by Myc transcription factors.

In conclusion this study contains data of very high technical quality and makes observations that are interesting for researchers in X chromosome inactivation. A few points should be addressed:

We thank the reviewer 2 for his/her comments on our work and helpful suggestions to improve the manuscript, all of which we answer below.

Major points

1.The authors investigate PcG marks that appear not strongly linked with silencing in X inactivation. However, macroH2A is not considered in the discussion that might have a role in maintaining repression at least temporarily. Could a statement be added if macroH2A histone variants are expressed and localized to the inactive chromosome.

As pointed out by the reviewer, MacroH2A is enriched on the inactive X chromosome (Okamoto et al, 2004). A discussion of the different marks, including MacroH2A, enriched on the inactive X has been included in our Introduction and Discussion.

Introduction, Pages 4-5, lines 88-90:

“The inactive X chromosome is also enriched for mono-methylation of histone H4 lysine K20, dimethylation of histone H3 lysine K9 and the histone variant macroH2A (Okamoto et al. 2004; Mak et al. 2004).”

Discussion, Pages 23-24, lines 530-539:

“Interestingly, expression of several epigenetic modifiers appeared to correlate with X-chromosome reactivation (Supplementary Table 3 and Supplementary Figure 2g) such as Kdm3a, Kdm3b and Kdm3c (Jumonji C domain-containing protein that demethylates for H3K9 methylation), but also Kdm2b (H3K36-specific demethylase). MacroH2A is enriched on the inactive X chromosome (Costanzi et al. 2000) and its variants (H2afy and H2afy2) are expressed in ICM cells (data not shown). MacroH2A might repress X-linked gene reactivation, in a redundant fashion with H3K27me3 marks or specifically for some genes (Gaspar-maia et al. 2013). Future work will be required to determine whether reactivation of the Xp in the ICM also requires erasure of other chromatin marks such as H3K9me2 or MacroH2A. Our findings open up the way for a better understanding of the in vivo requirements for epigenetic reprogramming in general.”

2. Page 13: On the important point of differentiating genes that reactivate early and late the conclusion seems not clear. Is reactivation correlated with some transcriptional property? From the Myc binding maybe promoter strength could be considered. Alternatively, chromatin organization and structure might be considered. Is there any idea which is a more likely mechanism or can one be ruled out?

In order to provide some insight into this important question concerning specific features/mechanisms that might be involved in the timing of reactivation of X-linked genes, we analysed for the different reactivation classes of X-linked genes, the genomic position along the X-chromosome, their location inside or in close proximity to first *Xist* entry sites, as well as their levels of expression (Supplementary Figure 2). No significant association between genomic localisation and reactivation timing has been found even if early reactivated genes have a tendency to be further from the *Xist* locus than late and very late reactivated genes (Supplementary 2b). We have modified the text accordingly.

Results, Pages 14-15, lines 318-329:

This is already quoted to address point 4 for reviewer 1.

Concerning promoter strength, we do not see that high level of expression is a predictor of early reactivation. This has now been included in the manuscript (Supplementary Figure 2).

To conclude, the only predictive feature that we could find for early-reactivated genes was a lack of enrichment of H3K27me3 repressive marks and a higher number of MYC binding sites.

Results, Pages 15-16, lines 344-349:

This is already quoted to address point 4 for reviewer 1.

3. Page 7 line 136: Loss of K27me in clusters of cells is interpreted as loss in one cell cycle could be active. This would be true if all cells cycle with the same rate. However, clustering could indicate that few cells replicate multiple times and loss of methylation could be still passive. Can this scenario be ruled out?

Cell division rate is not rapid in the inner cells mass. Previous work has shown that cell number increase rather slowly in the inner cell mass compared to trophoctoderm cells (Handyside and Hunter, 1986). Xp reactivation occurs in a time window of 6-12 hours suggesting that a maximum of one cell division has occurred.

Fig. 2. Cell numbers in blastocysts recovered at intervals on Days 4 and 5. Total (●), trophoctoderm (■) and inner cell mass (●) cells (mean \pm SD). (----) Best fit polynomial curve

Figure extracted from Handyside and Hunter, 1986

We thus believe that loss of H3K27me3 by a passive mechanism alone is unlikely.

However, in *Utx* mutant embryos, 20% of all

ICM cells still manage to lose their H3K27me3 enrichment. This could either be due to redundancy with other histone demethylase or a passive loss in some cells after few more hours.

4. Page 14 line 303 - 305: can this sentence be clarified: what are genes that are efficiently silenced? Is silencing not a prerequisite for being able to become reactivated later on?

We agree with the reviewer that our initial sentence was very confusing. Care has been taken to rephrase.

Results, Pages 14-15, lines 318-320:

This is already quoted to address point 4 for reviewer 1.

Minor points:

a) Abstract line 51: initial enrichment in K27me ... correlates with speed of reactivation. Should this be anticorrelated? or lack of enrichment?

Abstract has been corrected accordingly with “anticorrelated”. We thank the reviewer for pointing out this error.

b) Page 18 last sentence and title, abstract: .. different epigenetic memory states seems odd when the data suggest Myc TF binding as the clearest difference between early and late reactivators. It could be omitted from the title without losing the focus.

We propose a new title “Differential epigenetic landscapes and transcription factors explain X-linked gene behaviours during X-chromosome reactivation in the mouse inner cell mass”.

c) Figure 4A: The paternal expression of Xist is expected but of Ogt and YLpf6 is surprising. Can a brief explanation be added.

As remarked by the reviewer, *Ogt* and *Ylpf6* both display surprising paternal expression in the trophectoderm cells. Previous study from allele-specific scRNAseq, during pre-implantation development, in both parental origin crosses, has already highlighted specific behaviour of these two genes, classified as escapees in Borensztein *et al*, 2017. *Ogt* shows a trend for random monoallelic expression (Supplementary Figure 1) and *Ylpf6* is expressed from both chromosomes however it shows a strong bias for expression from the Castaneus allele according to Borensztein *et al*, 2017. In this manuscript for Nature Communications, all samples used for allele-specific information are from C57BL/6J x Castaneus hybrid cross, with the paternal X from Castaneus origin.

An explanation has now been included in the Figure legend for Figure 4a.

“As expected, Xist RNA is paternally expressed in the trophectoderm cells. Ogt and Yipf6 genes display similar paternal expression in the trophectoderm, escape imprinted XCI, and show random monoallelic expression and Castaneus bias respectively (Supplementary Figure 1) (Borensztein et al. 2017).”

Reviewer 3:

Preimplantation mammalian embryos feature a number of interesting global epigenetic processes and the dynamics of the X chromosome inactivation/re-activation is certainly one of them. The manuscript by Borensztein and colleagues provides a nice addition to our current (limited) knowledge about the process of Xp inactivation in the ICM of female mouse embryos. Using RNA FISH and scRNA Seq on cells isolated from E3.5-E4.0 mouse female embryos, the authors show that the reactivation of the Xp is more complicated than previously assumed. Early reactivating genes do not seem to be marked by H3K27me3 and their biallelic expression precedes the loss of Xist expression. To the contrary, late re-activating genes show enrichment of H3K27me3, the biallelic expression of these genes correlates with the loss of H3K27me3 and of Xist coating. Re-activation of these genes is also susceptible to the loss of Utx - H3K27me3 demethylase. Overall, this is a nice and interesting story that provides a number of interesting insights.

We would like to thank reviewer 3 for his comments on our work and his/her important suggestions to improve our manuscript.

I have a few comments that I think should be addressed:

1) Probably the most surprising finding is that some early re-activating genes appear to be biallelically expressed prior to the lineage segregation in the ICM. As the PrE cells are supposed to keep inactive Xp, the authors suggest that upon lineage segregation, these genes will be specifically re-silenced on the Xp in PrE cells. I wonder whether this is something that can actually be demonstrated? Transient erosion of the Xp could potentially lead to the epigenetic differences between the Xp in TE and PrE cells, is there any evidence for this?

The reviewer points out the potential epigenetic differences for the Xp between TE and PrE cells that might be due to transient reactivation, and we agree that this is a very interesting question. To our knowledge, no ChIPseq data of chromatin marks has been performed in PrE cells to compare them to TE cells, *in vivo*.

The fact that two early-reactivated genes, *Atrx* and *Abcb7*, both display biallelic expression in 90-100% of ICM cells at E3.5 based on RNA FISH (Figure 2), suggests that precursors of *both* Epi and PrE lineages show biallelic expression of those two genes. We have included a statement in

our manuscript.

Results, Page 13, lines 278-283:

“These data suggest that oscillations in the expression states of some genes on the Xp (such as Atrx) occur within a sub-population of ICM cells that will give rise to the PrE, where XpCI is known to be maintained, ultimately (West, Frels, and Chapman 1977). Our RNA-FISH data confirms that Atrx is transiently expressed from both X chromosomes even in the cells that will give rise to the PrE as it is found biallelically expressed in 90-100% of early ICM cells (Figure 2c).”

The authors claim that there is considerable transcriptional heterogeneity in the cells of the early ICM . However, looking at the PCA analysis, a possible interpretation might be that there is some separation already between the cells that might become future epiblast vs PrE. Can this interpretation be excluded?

We do agree with the reviewer that some degree of separation is already visible at E3.5 (Figure 3a). This is in line with previously published single cell microarray data (Oshini *et al*, 2014), which showed no separation at E3.25 but the first signs of separation at E3.5. However, it is still difficult to know which cells will give rise to which lineage at E3.5 according to well-known pluripotency and differentiation factors (Figure 3b and Figure 3c). Indeed, only a subset of E3.5 ICM cells clustered with either the PrE or Epi lineage cells (Figure 3c).

The manuscript has been modified accordingly.

Results, Page 11, lines 229-231:

“We found that E3.5 ICM cells still showed substantial heterogeneity compared to E4.0 ICM single cells, which clustered into two distinct groups. Nevertheless, some signs that 2 sub-populations are emerging could be seen at E3.5 for some ICM cells”

Results, Page 11, lines 238-241:

“No strong association was observed in E3.5 ICM cells with the exception of a few cells (n=3 potential pre-PrE and n=1 potential pre-Epi at E3.5, Figure 3b), supporting the idea that PrE and Epi lineages begin to be specified but are still not clearly established at the transcriptional level in E3.5 stage ICMs, as previously reported (Ohnishi et al. 2014).”

Also, if this was the case, can it be excluded that the biallelic expression of the early reactivating

genes can be seen only in the cells of early ICM that will eventually contribute only to the epiblast? (ie there is no transient biallelic expression in cells that will become PrE)?

As previously mentioned in our answer to the first question by reviewer 3, we believe that our RNA-FISH analysis of *Atrx* and *Abcb7* further supports their transient biallelic expression (Figure 2c). Indeed, both genes are found biallelically expressed in almost all the cells of the ICM (90 to 100%) at E3.5. Importantly, when lineage segregation is complete (E4.0 and onwards), the percentage of biallelic cells decrease for both *Atrx* and *Abcb7* to reach 50-60% of biallelic cells per ICM. This data, combined with the allele specific scRNAseq, suggests that both *Atrx* and *Abcb7* show transient biallelic expression in all cells – whether destined for PrE or Epi. We have adjusted the text as follows:

Results, Page 13- Results, Page 13, lines 278-283. This is already quoted to answer question 1 from reviewer 3.

2) Correlation is made between transcriptional reactivation of the early reactivating genes and *myc* (in reference to the reported hypetranscription in the ICM). I was wondering whether the authors performed an unbiased analysis of the TF binding site enrichment in the promoter regions of the early re-activating genes (vs late re-activating genes)? Why was *myc* specifically selected?

In the initial manuscript, we conducted a search of known TF binding site that might distinguish the different reactivation timing classes but no specific TF motif was found. We then decided to focus on potential candidates such as the pluripotency factors involved in ESC and epiblast formation (eg *Nanog*, *Oct4*, *Sox2*, *Esrrb*, *Myc*...). We analysed public ChIPseq data from Chen et al, 2008, for those candidates and only found specific enrichment of *Myc* binding sites in escapees and early-reactivated genes as reported in the manuscript.

Methods, Page 37

Transcription factor binding sites analysis

Nanog, *Oct4*, *Sox2*, *Myc*, *Mycn*, *Klf4*, *Esrrb* and *Tcfcp2l1* binding sites from ChIPseq experiments in mouse ESCs were taken from Chen et al., Mol Cell 2008(X. Chen et al. 2008). *Prdm14* binding sites in mESCs were taken from Ma et al., NSMB 2011(Ma et al. 2011). The number of binding sites of each factor in promoter, gene body and until 3kb upstream of the TSS was calculated for each gene of the reactivation-timing list (Supplementary Table 2).

As suggested by the reviewer we performed an unbiased analysis of TF binding sites, using the RSAT and then MEME Suite algorithms. This analysis highlighted the potential involvement of YY1 transcription factor in early reactivation and escapees. Interestingly, YY1 can act in synergy with MYC factors (Vella et al. 2012) and was recently linked to XCI escapees in humans (C.-Y. Chen et al. 2016). The Discussion and Methods sections have been modified accordingly.

Discussion, Pages 21-22, lines 489-500:

“In the search for other TFs potentially involved in early X-linked gene reactivation, we used algorithms for motif discovery (see Methods). No specific TF binding motif associated with escapees and early reactivated genes could be found with a high confidence. The lack of enrichment for known motifs could be due to the limited number of genes included in each of the reactivation classes. However, motif comparison analysis of any over-represented motifs in escapees and early-reactivated genes revealed a correspondence with the transcription factor YY1 (Ying Yang 1) (p -value=0.0002). This motif occurs 2.5 times more frequently in the group of escapees and early reactivated genes ($n=20$, 57 promoters), than in the group of late and very late genes ($n=7$, 49 promoters). YY1 is associated with escapees in human and has previously been described to be co-bound to the same binding sites as MYC in mouse ESCs (Vella et al. 2012; C.-Y. Chen et al. 2016). The precise roles of the MYC proteins and YY1 in relation to Xp gene activity merits future exploration.”

Methods, Pages 37-38

“Motif discovery analysis

RSAT oligo-analysis (Medina-rivera et al. 2015) was used to search for over-represented motifs in promoters (-700/+299nts relative to TSS) of X-linked genes in escapees, early, late and very late reactivation classes. Since the number of genes per class is too low to obtain high confidence results, we pooled genes by similar behaviour, with escapees and early reactivated genes in one group and late and very late in another one. One non-repetitive motif was found over-represented in the first group. This motif was compared to a database of known TF motifs using Tomtom (MEME Suite) (Bailey et al. 2009) and only one correspondence was found with E -value<1, that of the TF YY1 motif (p -value=0.0002, E -value=0.27, q -value=0.54). FIMO (MEME Suite) (Bailey et al. 2009) was used to determine the occurrences of this motif in each group of genes, and only matches with a p -value < 0.0001 were considered.”

3) What is the expression level of Utx in cells of early embryos (scRNASeq data)? Is there any difference between the cells of individual lineages? Are there any other candidate demethylases that could compensate for the loss of Utx?

We thank the reviewer for this comment. Level of expression of *Utx* has now been included in Supplementary Figure 3a. Interestingly, *Utx* is down-regulated in TE but stays high in ICM where Xp reactivation occurs.

Results, Page 18, lines 398-401:

“Interestingly Utx gene is expressed in pre-implantation embryos and remains high in early and mid ICM cells when it is down-regulated in trophectoderm (E3.5_TE), in which Xp inactivation is maintained (Supplementary Figure 3a).”

Another H3K27me3 demethylase that might potentially compensate for the loss of *Utx* in females (when *Uty* is absent) is JMJD3. A discussion of the different H3K27me3 demethylases and the choice of *Utx* as the best candidate is now included in the introduction.

Introduction, Pages 5-6, lines 110-119.

This is already quoted to address question 2 from reviewer 1.

Potential compensation of *Utx* by other demethylases is now included in the Discussion:

Discussion, Page 23, lines 515-518:

“The presence of some ICM cells with complete H3K27me3 erasure in Utx knock-out could be explain by compensation by other demethylases such as JMJD3 and/or by passive loss of the repressive mark during cell division, however very few cell divisions occur between E3.5 and E4.5 in ICMs (Handyside and Hunter 1986)”

4) Fig3: a)and b) there should be a distinct colour label for TE cells; there are seem to be more early E3.5 data points in the PCA analyses than in the hierarchical clustering (c) and d)). What is the reason for this?

Thanks to the reviewer’s suggestions, we have applied a distinct colour for TE cells, in order to clarify our PCA (Figures 3a and 3b).

Between PCA and hierarchical clustering, the exact same number of ICM cells has been used at E3.5. By PCA analysis on 1,000 most variable genes, E3.5 ICM cells all cluster together.

However, when we analysed the E3.5 ICM cells through pluripotency and lineage expression genes (Figure 3b, PCA, Figure 3c and 3d hierarchical clustering), some E3.5 ICM already clustered with either the PrE E4.0 cells (n=3) or the Epi E4.0 cells (n=1). This suggests that as early as E3.5, some cells are already specified toward a lineage.

Care has been taken to better explain these figures in the results section, as already quoted to answer question 1 (PCA) of the reviewer.

Results, Page 11.

Reviewer 4:

In this manuscript, Borensztein and colleagues performed a set of experiments to decipher the X-chromosome reactivation dynamics in the mouse embryonic development. Using a combination of RNA FISH and single-cell sequencing techniques, the authors discover that bi-allelic gene reactivation in the inner cell mass display different temporal dynamics that are, in part, linked with H3K27me3 epigenetic memory. The findings are topical and address factors that contribute to regulating the X-chromosome during development. In general, this study is well-designed and organized in a logical manner. However, several issues dampen my enthusiasm for the paper.

We thank reviewer 4 for his/her comments that strengthen our manuscript. Unfortunately we could not answer some of the requested changes (e.g. *Utx* scRNAseq to follow X-linked gene reactivation), which involve intensive animal work (e.g. new mutant strain on a Castaneus background) but we have tried to address them as far as we can in this revised version.

Major comments:

1) As shown in Figure 2, there are eight X-linked genes that show differential reactivation timing dynamics. The differential reactivation is not correlated with the linear distance from the Xist locus. Previous studies have shown that the spread of Xist RNA and gene repression is associated with the 3-dimensional architecture of the X-chromosome. The authors should cross-reference to such Xist-coating maps to determine any relationship between reactivation dynamics and Xist coating/H3K27me3 localization.

We acknowledge the reviewer for his/her remark. We analysed the first Xist “entry” sites reported in Engreitz *et al*, 2013 as being the first regions coated by Xist RNA during ESC differentiation. We also have previously shown that the first silenced genes during Xp inactivation *in vivo* seem to preferentially lie inside or next to these Xist entry sites (Borensztein *et al*, 2017). We have now compared the proportion of early, late and very late reactivated genes lying inside (TSS in the entry site), next to (TSS \pm 100bp) or outside a Xist entry site (Supplementary Figure 2c) and have found no differences between the various reactivation timing groups.

We also analysed the genomic location along the X chromosome and the level of expression of

the different classes of genes but did not find a significant predictive feature for early versus late reactivated genes (Supplementary Figure 2b and 2c).

Results, Pages 14-15, lines 318-329.

This is quoted to address point 4 for reviewer 1.

2) Related to the analysis of Single-cell RNA-sequencing, the authors should provide a table indicating the number of X-linked genes covered and the total number of SNPs that were used for each cell.

We now provide such information in our Supplemental table 1.

In addition, the single cell RNaseq protocol as described in this paper is well known to have 3' bias that could affect the SNP analysis. The authors should try an independent single-cell amplification approach, such as SMRT-seq2, which is capable of full length transcript profiling.

We agree with the reviewer that the Tang *et al* protocol has a 3' bias as the retro-transcription usually goes up to 3kb. However, in this analysis, we do not focus on different mRNA isoforms for which information is clearly lacking with our protocol. Rather we are interested the expression level and parental origin of our transcripts. In fact the 3'UTR region of the RNA is rich in polymorphisms and more than 60% of well-expressed genes are also informative in our study. Although a full-length scRNAseq protocol would have increased this percentage, we nevertheless compared our scRNAseq kinetics with nascent RNA Fluorescence in situ hybridisation of (RNA-FISH). This is a single cell, PCR-free technique that allowed us to confirm the timing of XCI for a subset of genes without any amplification bias.

We have changed our Methods section as follows:

Methods, Page 33:

“Estimation of gene expression levels.

RNA reverse transcription allowed sequencing only up to an average of 3 kb from the 3' UTR. To estimate transcript abundance, read counts were thus normalized on the basis of the amplification size of each transcript (retrotranscribed length per million mapped reads, RPRT) rather than on the basis of the size of each gene (RPKM), as described in Borensztein et al., 2017(Borensztein et al. 2017). To avoid noise due to single cell RNaseq amplification technique, only well-expressed genes (RPRT>4) were considered in our allele-specific study. A threshold of

RPRT>1 was applied to consider a gene as expressed (Figures 3, 4 and Supplementary Figures 3 and 4)."

The number of profiled TE cells (n=3) is also too few for proper statistical comparison given the large variation riddled in single-cell RNA-sequencing datasets. A subset of samples from male cells as a negative control would also be appropriate.

As requested, we have added some male TE and ICM cells from BC (BL/6J x Cast) and CB (Cast x BL/6J) cross for the Figure 4b. We also added two more TE female cells from the reverse cross CB to have a total of 5 TE cells for female and statistical analysis of correlation between gene expression and percentage of biallelic genes per cell (Figure 4b, Supplementary Table 3).

These samples were prepared under the same conditions.

Following on the reviewer's remark to add male as control, we consider that only genes expressed more than 1 RPRT are expressed in our study (Figure 4b, Supplementary Figure 2h).

3) As shown in Figure 3, the mouse single-cell RNA-seq data have previously been performed by Deng and Sandberg et al, the authors should use this orthogonal piece of data for cross-validating their findings.

We agree with the reviewer that the mouse allele-specific RNAseq data have been already produced by Deng *et al*, 2014 and we are citing them as reference 31 when comparing our kinetics of silencing for some gene candidates with their data.

However to our knowledge, they have not published any data from female mouse blastocysts at mid and late stages that would permit a comparison of our kinetics of X-linked gene reactivation. In Figure 3, we already corroborate findings from Ohnishi *et al*, 2014 (reference 34 in the manuscript), which are based on single cell microarray technique in mouse early, mid and late ICMs, and carefully following PrE and Epi segregation.

4) In the analysis of allele-specific expression, the authors cite their paper (now published in NSMB) that uses an analytical pipeline in which 15% is used as a cutoff for bi-allelic vs mono-allelic expression. Is this cut-off arbitrary, why not 10% or 5%?

The cutoff of 15% for biallelic expression was first arbitrarily chosen, based on Gendrel *et al*, 2015, and our previous paper, recently published in Nature Structural and Molecular Biology

(Borensztein *et al*, 2017). In the latter study, we validated the findings obtained by scRNAseq with previously published RNA-FISH studies. We were thus able to conclude that silencing kinetics could be accurately deduced for genes expressed more than 4 RPRT, with a cut-off of 15% for biallelic vs monoallelic expression.

From Borensztein *et al*, 2017, NSMB: “We compared the kinetics of Xp silencing for 13 X-linked genes previously analyzed by fluorescence *in situ* hybridization targeting nascent RNA (RNA-FISH) and found that most (12/13) genes showed very similar patterns (Fig. 1e and Supplementary Fig. 2), thus giving us confidence that our scRNA-seq data, bioinformatics pipeline and expression thresholds were valid.”

5) As shown in Figure 4, are there any transcriptome wide differences between sister cells from the same lineage that differ in % bi-allelic gene expression? In other words, does X-reactivation correlate with any other autosomal changes?

To answer this comment, we have performed a correlation analysis between gene expression (autosomes and X) and percentage of biallelic X-linked genes per cell. The following section has now been included in the Methods:

Methods, Pages 35-36:

“Correlation between autosomal and X-linked gene expression

Correlation and anti-correlation between gene expression levels (autosomes and X chromosomes) and percentage of X-linked gene reactivation (allelic ratio >0.2 for X-linked genes) was measured by Pearson correlation and Benjamini-Hochberg correction and are provided in Supplementary Table 3. BC and CB (only for E3.5 trophectoderm) female single cells have been used in this analysis.

Gene ontology has been made for the top correlated genes (q -value<0.05) with the Gene Ontology Project (Ashburner et al. 2000) and AMigo software (Carbon et al. 2009).”

Thanks to the reviewer’s important remark, this new analysis allowed us to associate X-chromosome reactivation with pluripotency factors on the one hand, such as Sox2 and Essrb or Nanog and Prdm14 that have already been hypothesized to play a role either in controlling *Xist* expression or during *in vivo* Xp reactivation (Navarro et al. 2008; Payer et al. 2013). On the other hand, X reactivation anti-correlates with differentiation factors involved in PrE formation such as Sox17, Gata6 and Gata4.

Moreover, a Gene Ontology analysis of the best-correlated genes (q -value <0.005) revealed an enrichment in epigenetic modifiers as might be expected in the context of epigenetic reprogramming. We have now added the following text:

Results, Page 16, lines 351-362:

“To explore the second hypothesis, that some TFs, including pluripotency factors, might drive expression from the Xp of a subset of early reactivated genes, we first analysed the correlation or anti-correlation between gene expression genome-wide and the degree of X-linked gene reactivation in female single cells (Supplementary Table 3, see Methods for details). As expected based on previous observations, Xp-chromosome reactivation correlates with pluripotency factors (e.g. Esrrb, Sox2, Nanog, Oct4 and Prdm14) and anti-correlates with PrE differentiation factors, such as Gata4, Sox17 and Gata6 (Okamoto et al. 2004; Payer et al. 2013; Mak et al. 2004; Navarro et al. 2008). A gene ontology analysis of the top correlated genes (q -values <0.005) revealed that epigenetic modifiers are overrepresented (Supplementary Figure 2g) and corroborates our hypothesis that different epigenetic landscapes might at least partially underlie the different reactivation kinetics.”

6) To examine the role of H3K27me3 in X-linked gene reactivation, the authors used the Utx mutant embryos and see some effects. It would be informative if single-cell sequencing data was available for comparison with Figure 4.

We agree with the reviewer that allele-specific scRNAseq of mutants would be very interesting. Unfortunately no polymorphisms are available with the *Utx* mutant strain however. We therefore cannot use RNA-seq to examine X reactivation. To do this would require creation of a new conditional *Utx* mutant on a *Mus musculus castaneus* background, which is clearly out of the scope of our study.

Finally, for the data analysis in Figure 5, the authors should correct for Type I error in the KW test as some of the p-values may become marginal.

We agree with the reviewer 4. We have performed a two-sided Dunn’s Multiple Comparison Test with Benjamini-Hochberg correction and Kruskal-Wallis analysis of variance.

P-values are still significant in Figure 5c, from * for p -value <0.05 to ** for p -value <0.001 .

Figures and legends have been accordingly modified and a statistical section is now added in

Methods.

Methods, Page 37:

“Statistics section

Kruskal-Wallis and Post-hoc test were used to analyse non-parametric and unrelated samples.

The statistical significance has been evaluated through two-sided Dunn's Multiple Comparison

Test with Benjamini-Hochberg correction and Kruskal-Wallis analysis of variance. p-values are

provided in the figures, figure legends and/or main text. Enrichment of histone marks has been

evaluated thanks to non-parametric Wilcoxon test.”

REVIEWERS' COMMENTS:

Reviewer #1 (Remarks to the Author):

The authors have generally suitably addressed reviewers' concerns. I have 2 (or 3) small issues to address:

(1) The revised title suggests to me that the reactivation kinetics are all explained, it seems to me to perhaps be more appropriate to include that these are: "Contribution of ...to reactivation kinetics?"

(2) On page 16/17 it is noted that approximately 1/2 of the X genes have binding sites – it would be important to know the frequency on autosomes – and I believe also whether this frequency is similar for all X-linked genes or only the ~20% examined in the study?

(3) It may be too awkward to include; however given that MacroH2A was included it seems that "erasure or modification of other marks" would be worth including (the or modification) as the authors' 2008 PNAS publication showed the importance of phosphorylation of MacroH2A for exclusion from the X chromosome.

Reviewer #2 (Remarks to the Author):

The revised version of the manuscript now entitled "Differential epigenetic landscapes and transcription factors explain X-linked gene behaviours during X-chromosome reactivation in the mouse inner cell mass" contains changes to the text and clarifications that have addressed my earlier criticism. In particular the authors have modified the title of the study, which is now reflecting the main emphasis of the work. In addition, the text has been extended to include explanations and clarifications that overall have considerably strengthened the manuscript. YY1 can now be shown to be associated with early reactivating genes in parallel with Myc TFs, and macroH2A is now discussed as an additional chromatin factor for silencing making the discussion of the results comprehensive and engaging. The resulting work will be of high interest to researchers in dosage compensation and early mouse development.

Minor points

a) Page 17, line 386 "thanks to" appears colloquial and could be rephrased to "by the". The sentence also appears long spanning the entire paragraph and not easy to read.

b) Page 17, line 392: "very-reactivated" is possibly erroneous or the meaning is not entirely clear.

c) Page 18, line 398: "Interestingly the Utx gene"

d) Page 23, line 527 "silencing state" could be changed to "silenced state"

e) Page 23, line 528: The sentence "..some genes resist full H3K27me3 during XCI ..." reads a bit odd

Reviewer #3 (Remarks to the Author):

The authors have addressed all my questions.

Reviewer #4 (Remarks to the Author):

In the revised manuscript, Heard and colleagues have significantly improved the overall writing as

well as provided additional data and interpretation. The authors have also sufficiently answered most of concerns, which has helped refine the interpretation of the study. I think this paper overall should be accepted in Nature Communications.

Point by Point Rebuttal for Borensztein, Okamoto et al

We would like to thank the reviewers for their enthusiastic comments and helpful suggestions. We have modified the revised manuscript to address all the points raised by reviewers 1 and 2.

Below we provide a point-by-point rebuttal to all of the comments made by the reviewers.

Reviewer 1:

We thank the reviewer 1 for his/her new suggestions to improve our manuscript.

The authors have generally suitably addressed reviewers' concerns. I have 2 (or 3) small issues to address:

(1) The revised title suggests to me that the reactivation kinetics are all explained, it seems to me to perhaps be more appropriate to include that these are: "Contribution of ...to reactivation kinetics?"

To answer both the reviewer's comment and the editorial request of 15 words maximum, we have changed the title for "Contribution of epigenetic landscapes and transcription factors to X-chromosome reactivation in the inner cell mass".

(2) On page 16/17 it is noted that approximately 1/2 of the X genes have binding sites – it would be important to know the frequency on autosomes – and I believe also whether this frequency is similar for all X-linked genes or only the ~20% examined in the study?

We are not sure exactly what value the reviewer was referring to here. Nevertheless, amongst the X-linked genes examined, we found a different frequency for MYC binding sites between the early-reactivated (31%) and escapee (42%) genes compared to the late (19%) and very late-reactivated (5%) genes.

To answer the reviewer's question concerning frequency on autosomes relative to the X, we extracted the TF-gene associated score provided by Chen et al, 2008, of the Myc factors not only for each gene in our reactivation-timing list, but also for all X chromosomal and autosomal genes. This has been added to the Methods section (Page 31) accordingly:

“Transcription factor binding sites analysis

Nanog, Oct4, Sox2, Myc, Mycn, Klf4, Esrrb and Tcfcp2l1 binding sites and their TF-gene associated score for each gene were taken from ChIPseq experiments in mouse ESCs, previously published in Chen et al., 2008 (GSE11431)⁴⁰. The TF-gene association score was calculated by Chen et al., Cell 2008⁴⁰, between each pair of gene and TF, from 0 to 1, with a higher score linked to a higher probability of the gene to be a direct target of the TF. For pluripotency factor analysis (Supplementary Fig. 3b), Nanog, Oct4, Sox2, Klf4, Esrrb and Tcfcp2l1 scores have been summed for each X-linked gene of the different reactivation-timing groups (Supplementary Data 2). In Supplementary Fig. 3c, Myc and Mycn scores have been summed. Mycl sites were not analysed in Chen et al⁴⁰.”

Thanks to this new analysis, we now show that early-reactivated genes and escapees are indeed enriched for their Myc-associated scores when compared to late and very late reactivated genes, but not to all X chromosomal and autosomal genes. It would therefore appear that the late and very late groups of reactivated X-linked genes are significantly depleted in Myc binding sites (p-value<0.0001, KW test).

Results section has been modified accordingly (Pages 15-16)

“We next examined previously published datasets of TF binding sites in mESCs and their TF-gene associated score as calculated by Chen et al, 2008⁴⁰ (see Methods). In particular we analysed the occurrence of fixation sites at X-linked genes for pluripotency factors involved in the Epiblast or mESC state (Nanog, Esrrb, Klf4, Oct4, Sox2, Tcfcp2l1). Half of the X-linked genes, independent of their kinetics of reactivation and including escapees, presented at least one binding site for these pluripotency factors (Supplementary Fig. 3a). Their expression might be partially regulated by these factors^{15,16}, but the binding of these factors alone cannot explain the behaviour of early reactivated genes. We next analysed Myc family binding sites, as Myc expression was also found associated with X-chromosome reactivation, though to a lesser degree than pluripotency factors (Supplementary Data 3): Myc factors are expressed in early and mid ICM cells and there is a slight but significant association between high expression of Myc and Mycl genes and high rate of X-linked gene reactivation (Supplementary Fig. 3b). We therefore analysed for the presence of Myc family binding sites (Myc and Mycn binding sites from Chen et al, 2008⁴⁰). Both escapees and early-reactivated genes showed a similar enrichment for Myc

factor binding sites compared to other X-linked and autosomal genes (Supplementary Fig. 3c). In comparison, late and very late reactivated genes were significantly depleted in Myc binding sites, when compared to early-reactivated genes, escapees but also genome-wide ($p < 0.0001$ by Kruskal-Wallis). Myc transcription factors play a role in iPS reprogramming⁴¹ and have also been linked with a hypertranscribed state, described in ESCs and Epiblast⁴². Thus, early-reactivated X-linked genes and escapees may well be more efficiently targeted for reactivation on the silenced paternal X by the Myc TF family in early ICM, compared to late and very late genes which have fewer Myc binding sites and would therefore be less responsive to these TFs.”

(3) It may be too awkward to include; however given that MacroH2A was included it seems that "erasure or modification of other marks" would be worth including (the or modification) as the authors' 2008 PNAS publication showed the importance of phosphorylation of MacroH2A for exclusion from the X chromosome.

We have modified our discussion accordingly.

Page 22, lines 578-580 :

“Future work will be required to determine whether reactivation of the Xp in the ICM also requires erasure or modification of other chromatin states such as H3K9me2 or MacroH2A.”

Reviewer 2:

We thank the reviewer 2 for his/her minor suggestions to improve the manuscript.

The revised version of the manuscript now entitled "Differential epigenetic landscapes and transcription factors explain X-linked gene behaviours during X-chromosome reactivation in the mouse inner cell mass" contains changes to the text and clarifications that have addressed my earlier criticism. In particular the authors have modified the title of the study, which is now reflecting the main emphasis of the work. In addition, the text has been extended to include explanations and clarifications that overall have considerably strengthened the manuscript. YY1 can now be shown to be associated with early reactivating genes in parallel with Myc TFs, and macroH2A is now discussed as an additional chromatin factor for silencing making the discussion of the results comprehensive and engaging. The resulting work will be of high interest

to researchers in dosage compensation and early mouse development.

Minor points

a) Page 17, line 386 "thanks to" appears colloquial and could be rephrased to "by the". The sentence also appears long spanning the entire paragraph and not easy to read.

To clarify our sentence, we have replaced “thanks to” according to the reviewer’s suggestion. We have also split the sentence into two.

Page 16, lines 379-385:

“In conclusion, the early reactivation of some X-linked genes, even prior to global loss of Xist RNA coating and H3K27me3 enrichment at E3.5, may be partly due to transcriptional activation by the Myc TF family, a local lack of H3K27me3 and an enrichment of H3K4me3. On the other hand, the majority of genes that are reactivated later show reduced numbers of Myc binding sites as well as higher H3K27me3 enrichment and lower H3K4me3. This suggests there may be differences between early and late reactivated genes in epigenetic memory states and responsiveness to some TFs.”

b) Page 17, line 392: "very-reactivated" is possibly erroneous or the meaning is not entirely clear.

We thank the reviewer for pointing out this error. We have now modified the sentence accordingly.

Page 16, lines 388-389 :

“The above findings (Figures 2 and 5) support a dependency between late and very-late reactivated genes on loss of Xist and H3K27me3 enrichment from the Xp.”

c) Page 18, line 398: "Interestingly the Utx gene"

We thank the reviewer for pointing out this error. We have now modified the sentence accordingly.

d) Page 23, line 527 "silencing state" could be changed to "silenced state"

We have now modified the sentence accordingly.

e) Page 23, line 528: The sentence "...some genes resist full H3K27me3 during XCI ..." reads a bit odd

We have clarified our sentence.

Page 22, lines 568-570:

"Furthermore, how and why certain genes appear to be excluded from H3K27me3 enrichment on the inactive during XCI and may thus be more prone to rapid reactivation, remains unknown."